



# A framework for assessing and understanding sources of error in Earth System Model emulation

Christopher B. Womack[1,2], Glenn Flierl[3], Shahine Bouabid[3], Andre N. Souza[3], Paolo Giani[2,3], Sebastian D. Eastham[4], and Noelle E. Selin[2,3,5]

[1]Department of Aeronautics and Astronautics, Massachusetts Institute of Technology, Cambridge, MA, United States
[2]Center for Sustainability Science and Strategy, Massachusetts Institute of Technology, Cambridge, MA, United States
[3]Department of Earth, Atmospheric, and Planetary Sciences, Massachusetts Institute of Technology, Cambridge, MA, United States
[4]Brahmal Vasudevan Institute for Sustainable Aviation, Department of Aeronautics, Imperial College London, London, United Kingdom
[5]Institute for Data, Systems, and Society, Massachusetts Institute of Technology, Cambridge, MA, United States

**Correspondence:** Christopher B. Womack (cwomack@mit.edu) and Noelle E. Selin (selin@mit.edu)

**Abstract.** Full-scale Earth system models are too computationally expensive to keep pace with the growing demand for climate projections across a large range of emissions pathways. Climate emulators, reduced-order models that reproduce the output of full-scale models, are poised to fill this niche. However, the large number of emulation techniques available and lack of a comprehensive theoretical basis to understand their relative strengths and weaknesses compromises fundamental methodolog-

ical comparisons. Here, we present a theoretical framework that connects disparate emulation techniques, using it to analyze sources of emulator error focusing on memory effects, hidden variables, system noise, and nonlinearities. This framework includes popular emulation techniques such as pattern scaling and response functions, relating them to less commonly used methods, such as Dynamic Mode Decomposition and the Fluctuation Dissipation Theorem (FDT). To support our theoretical contributions, we provide practical implementation details for each technique, evaluating performance across a series of

experiments designed to highlight different potential sources of error. We find that response function-based emulators outperform other techniques, particularly pattern scaling, across all scenarios tested. We additionally outline potential advantages of incorporating statistical mechanics into climate emulation through the use of the FDT, though this technique requires greater computational resources and non-standard scenarios for training. Results highlight the relative utility of each technique discussed, along with the importance of designing future scenarios for Earth system models with emulation in mind, suggesting

that large-ensemble experiments utilizing the FDT could benefit climate modeling and impacts communities.

## 1 Introduction

Earth-System Models (ESMs) are our most comprehensive tool to simulate the climate system, yet their high computational cost limits the range and number of scenarios that can be investigated (Flato, 2011; Müller et al., 2018). Growing demand for high-quality climate projections which differ from the scenarios considered within the Coupled Model Intercomparison

Project (CMIP) drives a need for computationally efficient alternatives (Eyring et al., 2016). Climate emulators - reduced order





models that reproduce the outputs of full-scale climate models - have seen a surge in popularity as they can be many orders of magnitude faster than the parent models (Sudakow et al., 2022; Tebaldi et al., 2025). Their low computational costs also make them an appealing tool to disseminate climate information to audiences beyond the climate science community.

Chaotic sensitivity renders prediction of the climate state infeasible beyond short time horizons (Lorenz, 2006, 2015). Climate emulators must therefore target the statistics of climate variables, such as means, variances, or higher moments, rather than simulating chaotic dynamics (Beusch et al., 2020; Souza et al., 2024; Wang et al., 2025). Many emulation techniques exist to estimate the mean state and/or probability distribution of climate variables (Meinshausen et al., 2011; Castruccio et al., 2014; Herger et al., 2015; Tebaldi and Knutti, 2018; Leach et al., 2021; Watson-Parris et al., 2022; Addison et al., 2024; Bassetti et al., 2024; Bouabid et al., 2024), and in this work we explore methods that emulate the mean state of the system. In a recent review, Tebaldi et al. (2025) distinguished between five main categories of climate emulators, including linear pattern scaling, statistical approaches, and machine learning algorithms. Following their categorization, we focus on linear pattern scaling and its immediate extensions along with dynamical system/impulse response theory emulators.

In the climate context, the most commonly used emulation technique is pattern scaling (Santer et al., 1990), a simple linear regression of local climate variables (e.g. temperature or precipitation anomaly) on the global mean temperature anomaly. Pattern scaling has been used and studied extensively since its development (Mitchell, 2003; Tebaldi and Arblaster, 2014; Wells et al., 2023; Giani et al., 2024), with variations that capture seasonal anomalies, different mixes of greenhouse-gases, and spatially heterogeneous forcings such as aerosols (Schlesinger et al., 2000; Herger et al., 2015; Mathison et al., 2024). This approach produces accurate projections assuming exponential and fixed-pattern forcing, linear feedbacks, and linear and time-independent dynamics, criteria that are roughly satisfied in a number of CMIP experiments (Giani et al., 2024). Memory effects in overshoot scenarios (forcing history, rather than only instantaneous forcing, affecting a future state) violate these assumptions, causing this approach to break down for many decision-relevant scenarios.

Impulse response (response/Green's function) methods fill this memory effect gap by encoding forcing history into the emulator, rather than relying only on the instantaneous forcing. These techniques have been studied thoroughly in the contexts of dynamical systems and climate science (Joos and Bruno, 1996; Hasselmann et al., 1997; Lucarini et al., 2017; Orbe et al., 2018; Freese et al., 2024; Giorgini et al., 2024), and are an active area of research (Winkler and Sierra, 2025). Response functions are popular due to their ease of interpretability and improvement in skill over pattern scaling in capturing realistic dynamics (Womack et al., 2025). Pure linear response functions cannot account for nonlinear effects, though hybrid schemes that incorporate machine learning (ML) may help resolve this issue (Winkler and Sierra, 2025).

Pattern scaling and linear response functions are prevalent in climate emulation literature, yet these approaches are only two methods among a broad spectrum of emulators, with each technique offering trade-offs in terms of complexity, data requirements, and interpretability. For example, quasi-equilibrium emulation is closely related to pattern scaling, though only a handful of studies explore the utility of this principal beyond the traditional choice of global mean temperature as emulator input (Huntingford and Cox, 2000; Cao et al., 2015). Other techniques, such as Dynamic Mode Decomposition (DMD) and its variants, are generally not classified as emulators despite their potential to identify and predict modes of variability in the climate system (Kutz et al., 2016; Gottwald and Gugole, 2020; Navarra et al., 2021; Mankovich et al., 2025).





We consider climate emulators as defined in Tebaldi et al. (2025), excluding Simple Climate Models (SCMs) and Earth system Models of Intermediate Complexity (EMICs), though they share similarities with emulators. We also do not examine ML emulators such as FourCastNet and NeuralGCM – while these techniques are promising for weather prediction, they currently lack the stability required for reliable climate prediction (Pathak et al., 2022; Kochkov et al., 2024). Several studies have employed ML techniques to instead target the statistics of the climate, rather than weather (Lewis et al., 2017; Bassetti et al., 2024; Wang et al., 2025), but these works focus on emulator implementation rather than theoretical analysis.

In this work, we develop a framework connecting a spectrum of emulators through the Koopman and Fokker-Planck operators, which govern the evolution of stochastic processes. In doing so, we identify a gap in the Tebaldi et al. (2025) emulator typology: operator-based emulators, an area largely unexplored in existing climate emulator literature. While previous work has connected operator frameworks with the Fluctuation Dissipation Theorem and thus, linear response theory (Cooper and Haynes, 2011; Lucarini et al., 2017; Lembo et al., 2020; Zagli et al., 2024; Giorgini et al., 2025), our contribution explicitly demonstrates its utility in the context of climate emulation. Section 2 first presents our theoretical framework (Sect. 2.1), highlighting that emulators attempt to simplify complex climate dynamics into a linear set of modes associated with the Fokker-Planck and Koopman operators. We then apply this framework to identify potential sources of emulator error (Sect. 2.2), comparing six emulation techniques from both a theoretical and practical perspective, providing a toolkit for those interested in emulating dynamical systems. We then introduce a series of experiments using simplified climate models and forcing scenarios designed to stress test and evaluate each emulator (Sect. 2.3). Section 3 contains experimental results, showing that response functions consistently outperform other emulators across potential high-error scenarios. We conclude by discussing optimal use cases for each emulator in Sect. 4.

## 2 Methods

Section 2.1 outlines a theoretical framework for analyzing emulators, while Sect. 2.2 provides theory and implementation details for the six emulators of interest. We detail experiments and emulator evaluation protocol in Sect. 2.3. See Fig. 2 for a conceptual roadmap of emulator theory and Table 1 for an overview of selected methods.

Throughout this section, we denote scalars with lowercase characters, vectors with lowercase, boldface italic characters, matrices with uppercase, boldface characters, and operators with script characters (e.g. $\mathcal{N}$ or $\mathcal{L}$). We use $\boldsymbol{x}$ and $n_x$ to denote the spatial coordinate and its dimensionality, along with $t$ and $n_t$ to denote the temporal coordinate and its dimensionality. Our examples focus on climate anomalies relative to a background state, though these techniques are applicable to general chaotic dynamical systems.

### 2.1 Theoretical framework for climate emulation

Section 2.1.1 introduces our emulation target, a general, stochastic system, outlining potential sources of error when emulating this system. Section 2.1.2 formalizes two complementary emulation strategies: emulating the full probability distribution, or emulating a collection of statistical moments (e.g. mean, variance).





### 2.1.1 Problem setup

A full-scale climate model is a deterministic, albeit chaotic, system. This chaos results in extreme sensitivity to initial con-
ditions, requiring emulation of the system's statistics, rather than its dynamics (Lorenz, 2015). To understand the statistics of
the system and how they may change over time, we follow Hasselmann (1976) in modeling the evolution of a single climate
variable using a stochastic differential equation (SDE) (Fig. 2, box 1). We treat short-term weather fluctuations as stochastic
noise to parameterize their influence on the long term, chaotic behavior of the system

$$\frac{\partial w}{\partial t} = \mathcal{N}(w) + F(t) + \varepsilon\xi(t), \tag{1}$$

where $w$ is the climate variable (or set of variables) of interest, $F$ is an external forcing (e.g. $CO_2$), $\mathcal{N}$ is the operator governing
the evolution of that variable, $\xi$ is a white noise term, and $\varepsilon$ is the noise standard deviation. We consider variables of interest
to be anomalies relative to some base state (e.g. temperature anomaly with respect to preindustrial conditions). $\mathcal{N}$ may involve
both linear and nonlinear terms in one or several fields, and we cannot directly represent this operator; this parameterization
aggregates the effects of processes such as heat and momentum transfers. The operator may also be influenced by variables
we observe as well as unobserved hidden variables (e.g. aerosol forcing in a pattern scaling emulator with only global mean
temperature as an input). The noise standard deviation can also be state dependent, though we treat it as independent for this
exploration.

Climate emulators approximate Equation 1, either implicitly (pattern scaling) or explicitly (Dynamic Mode Decomposition),
rendering them vulnerable to several potential sources of error. Figure 1 provides an overview of the sources of error we
consider across a range of scenarios: Errors can enter from the forcing if an emulator assumes only the instantaneous forcing is
significant and not the forcing history (Fig. 1 (a) - memory effects in an overshoot scenario). The presence of hidden variables
can lead to errors in some techniques (Fig. 1 (b) - localized aerosol effects when assuming well-mixed forcings), while other
techniques are sensitive to noise (Fig. 1 (c) - overfitting on internal variability). Finally, any linear emulation technique will
break down in the presence of nonlinearities (Fig. 1 (d) - ice-albedo feedbacks).

### 2.1.2 Operator framework for emulators

Our operator framework simplifies complex, possibly nonlinear climate dynamics into a linear set of modes with associated
decay rates. We use the term operator to refer to an update rule that advances the system one timestep for a quantity of
interest. An emulator attempts to approximate these modes, which are physically interpretable; for temperature, the decay rates
correspond to heat-uptake timescales.
Table 1 summarizes emulation techniques discussed in this section, providing a short conceptual description of each method
along with their key assumptions. We focus on linear emulation techniques that target the mean state of a climate variable:
pattern scaling, the Fluctuation Dissipation Theorem (FDT), deconvolution, modal fitting, Dynamic Mode Decomposition
(DMD), and Extended DMD (EDMD). The FDT, deconvolution, and modal fitting emulators are all response function-based
emulators, while E/DMD are operator-based emulators.





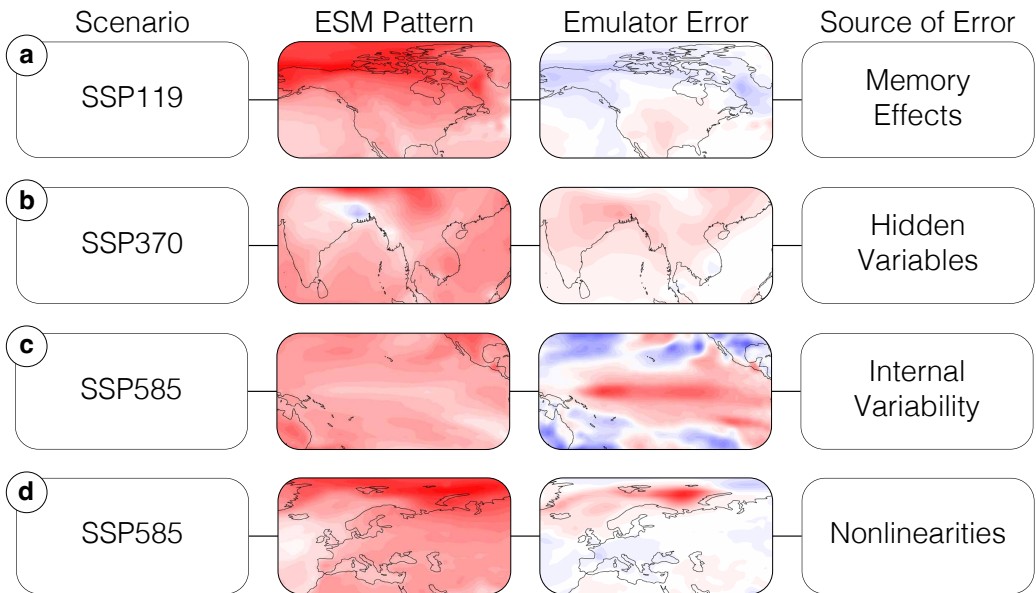

**Figure 1.** Potential sources of emulator error by scenario; emulator errors shown here are meant for illustrative purposes only. (a) Pattern scaling emulator trained on *historical* and *SSP585*, tested against *SSP119* in 2100; error over northern North America results from memory effects. (b) Pattern scaling emulator trained on *historical*, tested against *SSP370* in 2050; error over India and SE Asia results from hidden variables (aerosols not contained in training data). (c) High-order polynomial pattern scaling emulator trained on *historical*, tested against *SSP585* in 2020; error results from overfitting on internal variability. (d) Pattern scaling emulator trained on *historical*, tested against *SSP585* in 2100; error results from nonlinear feedbacks in the Arctic. All ScenarioMIP data shown are taken from the MPI Grand Ensemble (O'Neill et al., 2016; Maher et al., 2019).

**Emulating a probability distribution.** Our governing system, Equation 1, simulates a variable of interest, $w$, forward in time under a stochastic forcing. The trajectory of the time evolution of $w$ is characterized by the probability distribution, $p(w,t)$. We therefore focus our efforts on emulating $p(w,t)$ via the Fokker-Planck operator. This is a mathematical tool to evolve the probability distribution of a stochastic system forward in time. As this operator is linear, emulating it is equivalent to approximating a series of eigenvalues and eigenfunctions.

As shown by Hasselmann (1976), the time evolution of $p(w,t)$ is given by the Fokker-Planck equation corresponding to the governing SDE

$$\frac{\partial}{\partial t}p(w,t) = -\frac{\partial}{\partial w}\left[p(w,t)\left(\mathcal{N}(w) + F(t)\right)\right] + D\frac{\partial^2}{\partial w^2}p(w,t), \tag{2}$$

where $D$ is a diffusion coefficient set by the noise term, $D = \varepsilon^2/2$. The Fokker-Planck equation describes how the probability density evolves in time and can be viewed as an advection-diffusion process.

Advection, which shifts the mean of $p(w,t)$, occurs due to the deterministic action of the governing operator and the external forcing. Because the advective term acts on the flux, it both shifts the mean and reshapes the density. Diffusion, which increases



**Figure 2.** Conceptual flowchart for building an emulator through the joint Fokker-Planck/Koopman operator framework. Pop-outs show specific emulation techniques, while the shaded color indicates which concept a class of emulators relates to. Dashed arrows indicate conceptual/theoretical connections and solid arrows indicate a direct pathway. The overall process is as follows: (1) Select a climate variable of interest, $w$, such as temperature, here parameterized as the output of a stochastic differential equation. (2) Choose an emulation target, either the full probability distribution (option 1; 2a, 3a, 3c) or a statistical quantity such as the mean or variance (option 2; 2b, 3b, 3d). (3) Construct an emulator by selecting an approximation for either the Fokker-Planck or Koopman operator, including their response function representations; these options are connected through duality and are directly linked to linear response theory. (4) Given a new scenario of interest, emulate selected variable.

the variance in $p(w, t)$, is driven by system noise. Integrating Equation 2 forward diffuses the probability distribution, initially increasing the variance of $w$ until balanced by the mean-reverting drift ($\mathcal{N}(w) + F(t)$). It is common practice to write a



**Table 1.** Summary of emulation techniques discussed in this work including a short description and their key assumptions. Fluctuation Dissipation Theorem assumptions are shared with deconvolution and modal fitting emulation techniques. All techniques except the Fluctuation Dissipation Theorem additionally assume no hidden variables.

| Technique | Short Description | Key Assumptions |
|---|---|---|
| Method I: Pattern Scaling | Time-invariant pattern based on global mean temperature | Climate is always near equilibrium; response is instantaneous; fixed spatial pattern |
| Method II: Fluctuation Dissipation Theorem | Response functions derived through perturbation ensemble experiments | Perturbations are small; data come from linear response regime |
| Method III: Deconvolution | Response functions solved for from any general experiment | Quasi-equilibrium initial condition; influence of noise is small |
| Method IV: Modal Fitting | Response functions fit from any general experiment | Response is a decaying exponential; few significant modes |
| Method V: Dynamic Mode Decomposition (DMD) | Approximating system dynamics with a linear operator | Dynamics are approx. linear; training data capture relevant dynamics |
| Method VI: Extended DMD | Approximating system dynamics with nonlinear basis functions | Basis functions span Koopman operator; dynamics are approx. linear in new basis |

Fokker-Planck equation directly from an SDE, as there exists a general relationship between any SDE and its corresponding
Fokker-Planck equation; the full general derivation can be found in Denisov et al. (2009).

Importantly, the right hand side of Equation 2 is linear in the derivatives of $w$, allowing us to rewrite it in terms of the linear Fokker-Planck operator, $\mathcal{F}$,

$$\mathcal{F}(\cdot) = \frac{\partial}{\partial w}\left[D\frac{\partial}{\partial w}(\cdot) - (\cdot)(\mathcal{N}(w) + F(t))\right], \tag{3}$$

where the notation $\mathcal{F}(\cdot)$ means the Fokker-Planck operator is acting on some arbitrary variable (in our case, $p(w,t)$ in Equation
2). The Fokker-Planck operator (Fig. 2, box 3a) gives us a linear method to represent the time evolution of the probability distribution. Linearity additionally allows us to decompose $\mathcal{F}$ into eigenvalues and eigenfunctions (continuous eigenvectors). These are the target of our emulator, and our emulator skill is directly proportional to how well it can approximate those eigenvalues and eigenfunctions, along with our estimate of $p(w,0)$. This eigendecomposition is given by

$$\mathcal{F}f_{\mathcal{F}} = \lambda_{\mathcal{F}}f_{\mathcal{F}}, \tag{4}$$



where $\lambda_{\mathcal{F}}$ denotes an eigenvalue and $f_{\mathcal{F}}$ denotes an eigenfunction of the Fokker-Planck operator. The collection of $\lambda_{\mathcal{F}}$ and $f_{\mathcal{F}}$ fully characterizes the system's behavior. Our stochastic system evolves as a linear combination of probability distributions, $f_{\mathcal{F}}$, each decaying at rate $\lambda_{\mathcal{F}}$; the real part of the eigenvalues controls the decay rate, while any imaginary components result in oscillations over time. In the advection-diffusion analogy, each eigenfunction is a probability parcel that is carried and spread by the flow. The imaginary parts of the eigenvalues transport this parcel (shifting the mean) while the real parts act like an

effective diffusivity (increasing the variance). This tells us which physical behaviors dominate and on what timescales they matter for climate prediction.

Unfortunately, in most cases we cannot obtain an explicit representation of the Fokker-Planck operator due to $\mathcal{N}$ being nonlinear; see Appendix C for an analytic example of when this is possible. Because it acts on functions, the operator is infinite dimensional with infinitely many eigenpairs. This poses an immediate issue since computers have a finite amount of

memory. Finite dimensional matrix approximations of the Fokker-Planck operator have been studied (often framed through the more general Perron-Frobenius operator) (Klus et al., 2016, 2018; Kaiser et al., 2019; Souza, 2024b, a; Souza and Silvestri, 2024), but require a large amount of data to reliably estimate the operator. For climate emulation this poses an additional issue, as generating large enough ensembles to resolve $p(w, t)$ is prohibitively expensive. Because of these difficulties, little work exists studying the Fokker-Planck/Perron-Frobenius operator in the climate context (Navarra et al., 2021), though methods that

reconstruct the full probability distribution of a climate variable using statistical methods (e.g. diffusion models and Gaussian processes) implicitly represent it (Bassetti et al., 2024; Bouabid et al., 2024; Wang et al., 2025).

**Emulating a statistical quantity.** In practice, it is often easier to emulate statistical quantities, such as the mean or variance of a climate variable. Many common emulation techniques (e.g. pattern scaling and response functions) target only the mean of a single variable (Herger et al., 2015; Wells et al., 2023; Freese et al., 2024), though other work extends this to approximate

second order moments (Beusch et al., 2020; Wang et al., 2025). Relating these techniques requires the use of Koopman operator theory (Fig. 2, box 3b), a linear framework for propagating statistical quantities (usually referred to in the Koopman literature as statistical observables) forward in time (Mezić, 2013; Otto and Rowley, 2021). Emulator studies rarely link their methods to Koopman theory, while literature that explicitly connects to the theory does not use the same emulator terminology (Slawinska et al., 2017; Navarra et al., 2021), though they accomplish similar prediction tasks. The Koopman operator allows for an

exact representation of nonlinear dynamics using a linear operator, making it appealing when studying complex systems. We show how it can be used to emulate climate variables, simplifying nonlinear processes to the linear problem of emulating physically-interpretable eigenvalues and eigenfunctions.

To derive the Koopman operator, we first define a general statistical quantity, $g(w)$, whose expectation, $\langle \cdot \rangle$, is given by

$$\langle g(w) \rangle = \int g(w) p(w, t) \, dw, \tag{5}$$

We then take the time derivative of this expression, moving the partial derivative inside the integral to act only on $p$ since $g(w)$ is independent of time. This allows us to substitute the resulting expression into the right hand side of Equation 2. Integrating this by parts twice gives

$$\frac{\partial}{\partial t} \langle g(w) \rangle = \left\langle \left[ \mathcal{N}(w) + F(t) \right] \frac{\partial}{\partial w} g(w) \right\rangle + D \left\langle \frac{\partial^2}{\partial w^2} g(w) \right\rangle, \tag{6}$$





where the diffusivity, $D = \varepsilon^2/2$, is identical to the Fokker-Planck case. This form allows us to define the Koopman operator,

$\mathcal{K}$. It is linear in its derivatives of $w$, and we rewrite it as

$$\mathcal{K}(\cdot) = \mathcal{N}(w)\frac{\partial(\cdot)}{\partial w} + D\frac{\partial^2(\cdot)}{\partial w^2}, \tag{7}$$

where the notation $\mathcal{K}(\cdot)$ means the Koopman operator is acting on some arbitrary variable ($g(w)$ in Equation 7). Substituting this into Equation 6 gives

$$\frac{\partial}{\partial t}\langle g(w)\rangle = \langle \mathcal{K}g(w)\rangle + F(t)\left\langle \frac{\partial}{\partial w}g(w)\right\rangle, \tag{8}$$

This expression applies to any arbitrary statistical quantity (of which there are infinitely many), thus it can be used to integrate every statistical quantity forward in time; it is an alternate way to represent the complete probability distribution by representing each individual statistic. A useful choice is to select $g(w) = w$, giving

$$\frac{\partial}{\partial t}\langle w\rangle = \langle \mathcal{K}w\rangle + F(t), \tag{9}$$

which we will refer back to later.

Analogously to the Fokker-Planck operator, the Koopman operator provides a linear method to represent the time evolution of our entire collection of statistical quantities. As before, we can perform an eigendecomposition on the Koopman operator

$$\mathcal{K}f_\mathcal{K} = \lambda_\mathcal{K}f_\mathcal{K}, \tag{10}$$

where $\lambda_\mathcal{K}$ denotes an eigenvalue and $f_\mathcal{K}$ denotes an eigenfunction. The time evolution of our statistical quantity of interest is a linear combination of these eigenpairs. These can be used to identify dominant system dynamics and on what timescales
they emerge. Training an emulator is equivalent to approximating eigenpairs; reproducing these pairs accurately emulates the behavior of the system.

    However, approximations of the Koopman operator are limited by the same finite memory constraint as the Fokker-Planck case and deriving analytic solutions is dependent on the exact form of $\mathcal{N}$; see Appendix C for an example of when analytic approximations are possible. Matrix approximations of the Koopman operator are nevertheless more prevalent than their
Fokker-Planck counterparts (Schmid, 2010; Mezić, 2013; Williams et al., 2015; Otto and Rowley, 2021). Variants of these methods have recently been implemented in the climate context to identify dominant modes of variability in the system (e.g. El Niño-Southern Oscillation or Pacific decadal oscillation) (Navarra et al., 2021, 2024; Mankovich et al., 2025), but have not been applied for the purpose of climate emulation. We outline two of these methods explicitly in Sect. 2.2.3.

    **Two sides of the same coin.** The Koopman operator advances all statistical quantities of interest, and provides an alternative
to the Fokker-Planck description of a distribution's time evolution. Knowing every statistic is equivalent to knowing the full distribution. Access to either operator fully characterizes our system, allowing us to emulate it. Mathematically, these operators are dual (adjoint), where duality refers to two mathematical objects that contain alternate descriptions of the same information; this property is how we derived the Koopman operator in the previous section. This is analogous to, but physically and mathematically distinct from adjoint methods in climate modeling. There, adjoints to dynamics (rather than statistics as is the case for





the Koopman/Fokker-Planck approach) are exploited to calculate gradients with respect to input parameters more efficiently, which can be used to tune parameters and compute output sensitivities (Thuburn, 2005; Henze et al., 2007; Lyu et al., 2018).

Due to internal variability in the climate system, estimating the full probability distribution of a variable requires large initial condition ensembles, incurring significant computational cost. This is exacerbated for variables such as precipitation, where internal variability masks the forced response to a greater degree (Blanusa et al., 2023). Reliably estimating the full

distribution at each timestep to approximate the Fokker-Planck operator from relatively coarse data is impractical. However, under additional assumptions of quasi-ergodicity, we bolster our sampling power by assuming that the statistics do not change sufficiently quickly over a given time period. We thus focus on emulating lower-order statistical quantities, presenting those techniques in Sect. 2.2.

**Connecting to linear response theory.** Linear response theory states that the climate system's forced response (assuming

perturbations are small) is encoded by a response function, $R(t)$. The response function is generated by the Koopman operator, $\mathcal{K}$, where each eigenpair of the operator determines the characteristic timescales of the system. Considering temperature anomaly as an example variable, fast modes map to rapid atmospheric adjustments, while slow modes capture deep ocean heat uptake (Caldeira and Myhrvold, 2013). Response functions have been applied to a variety of climate problems (Joos and Bruno, 1996; Hasselmann et al., 2003; Joos et al., 2013; Orbe et al., 2018; Cimoli et al., 2023), including climate emulation

(Freese et al., 2024; Womack et al., 2025; Sandstad et al., 2025), though often without addressing the formal response theory underlying these techniques. As was the case with the Koopman operator, more formal applications of response theory to climate science often do not share the same language as climate emulators despite the shared goal of predicting the climate's forced response (Lucarini et al., 2017; Lembo et al., 2020; Zagli et al., 2024).

Making the relationship between response theory and the Koopman operator explicit in the context of emulation requires

the use of the Fluctuation Dissipation Theorem (FDT) (Lucarini et al., 2025). The FDT describes how a system (e.g. the Earth system) responds to perturbations (anthropogenic $CO_2$ emissions) relative to some baseline state (preindustrial conditions). Formally, it states that the response function associated with the statistical field $g$ is calculated by computing the temporal autocorrelation with the system's score function, $s$,

$$R(t) = \langle g(t = t')s(t' = 0)\rangle, \tag{11}$$

where the score function of the steady-state distribution encodes how a small perturbation alters the system's statistics; see Giorgini et al. (2024, 2025) for more details. The change in the ensemble average field, $\delta g$, is then obtained by convolving with a forcing, $F$,

$$\langle \delta g \rangle = \int\limits_{-\infty}^{t} R(s)F(t-s)\,ds. \tag{12}$$

The connection to Koopman operator theory is that temporal autocorrelations are expressed explicitly in terms of the Koopman

operator, see (Zagli et al., 2024).

Equation 12 is one way to state the Fluctuation Dissipation Theorem (FDT, Fig. 2, box 3d), a tool widely used in statistical mechanics and one of the main features of linear response theory (Lucarini et al., 2017; Lembo et al., 2020). The FDT predicts





the first-order response of a statistical quantity due to external perturbations and is defined in terms of an ensemble average over a quantity of interest. As written, this form does not account for state- or time-dependent effects (i.e. one could consider the alternate formulation: $R = R(w, t, t')$), though extensions to capture these effects and higher order statistical moments have been proposed (Metzler et al., 2018; Giorgini et al., 2025; Winkler and Sierra, 2025).

Response function emulators approximate the left hand side of Equation 11 using a variety of techniques, which we outline in more detail in Sect. 2.2.2. Their emulation goal is typically either to fit the eigenpairs which make up $\mathcal{K}$ explicitly (Sandstad et al., 2025), or to find a direct representation of $R(t)$ (i.e. an implicit representation of $\mathcal{K}$) (Lembo et al., 2020; Freese et al., 2024; Womack et al., 2025). The former may be more easily interpretable through analyzing the explicit eigenpairs, while the latter offers flexibility in allowing for parametric forms other than a decaying exponential.

Response theory builds upon the operator frameworks presented in the previous sections by providing a method to illustrate how a given quantity responds to small changes in forcing. While the Fokker-Planck and Koopman perspectives offer complete characterizations of the statistics of the system over time, response theory offers a practical approach to use this information to predict how a quantity shifts under perturbations, described by the FDT.

## 2.2 Emulation under this framework

Following the framework from the previous section, we introduce several emulation techniques targeting the mean of a climate variable (Fig. 2, pop-outs on right hand side). We use the example of estimating the expected (or annual-average) temperature anomaly, $T(\boldsymbol{x}, t)$, given an external forcing, $F(t)$ (e.g. $CO_2$ or other GHG emissions), though these techniques can be applied to any climate field. Each technique relates explicitly to the Fokker-Planck or Koopman operator and/or the Fluctuation Dissipation Theorem (FDT). We begin with methods that impose strong assumptions on the underlying data and progressively lift those assumptions until we are left with the most general emulation techniques; headings follow the taxonomy of Tebaldi et al. (2025) when possible.

### 2.2.1 Pattern scaling and its immediate extensions

**Method I: Pattern Scaling.** Pattern scaling is arguably the most well-known climate emulation technique (Santer et al., 1990; Mitchell, 2003; Tebaldi and Arblaster, 2014; Kravitz et al., 2017; Tebaldi and Knutti, 2018; Wells et al., 2023; Giani et al., 2024); it is formally derived via the Koopman operator, and is a specific case of a more general quasi-equilibrium emulation framework. It assumes that, at any given moment, the climate is in a quasi-equilibrium, rather than a transient, state and that changes in the forcing are small enough and/or the response of the system is fast enough to neglect system memory. Pattern scaling also assumes that the response does not depend on the background climate state, only the instantaneous forcing. Despite work showing that there are measurable differences between transient and quasi-equilibrium climate responses depending on the transient warming rate (King et al., 2021), the success of pattern scaling has led to its continued use.

We first restate Equation 9 in terms of the quasi-equilibrium assumption and our climate variable of interest as

$$\frac{\partial}{\partial t} T(\boldsymbol{x}, t) = \mathcal{L}(\boldsymbol{x}, \boldsymbol{x}') T(\boldsymbol{x}', t) + F(t) \approx 0, \tag{13}$$



where $\mathcal{L}$ indicates that this is no longer the true Koopman operator and $\boldsymbol{x}$ and $\boldsymbol{x}'$ indicate summation over spatial interactions, i.e. how one location, $\boldsymbol{x}$, is influenced by all other locations (including itself), $\boldsymbol{x}'$; a more detailed description of the transition from Equation 9 to 13 can be found in Appendix A4. We additionally assume $T(\boldsymbol{x}, t)$ here refers to the ensemble mean temperature, which has the practical advantage of reducing the impact of internal variability on our emulator. Inverting this equation gives

$$T(\boldsymbol{x}, t) = -\mathcal{L}^{-1}(\boldsymbol{x}, \boldsymbol{x}')F(t), \tag{14}$$

which is a more general formulation of pattern scaling based on a generic forcing, $F(t)$. Alternate definitions of pattern scaling have been explored previously, with a handful of studies developing extensions based on alternatives to global mean temperature such as radiative forcing or a combination of factors (Huntingford and Cox, 2000; Cao et al., 2015). A traditional pattern scaling formulation makes the further assumption that the forcing is the global mean temperature anomaly, $F(t) = \overline{T}(t)$,

and replaces $\mathcal{L}^{-1}$ with a low-order polynomial, leading to

$$T(\boldsymbol{x}, t) = a_0(\boldsymbol{x}) + a_1(\boldsymbol{x})\overline{T}(t) + \frac{1}{2}a_2(\boldsymbol{x})\overline{T}^2(t) + \dots, \tag{15}$$

where $a_i(\boldsymbol{x})$ indicates the spatially varying pattern, and we typically keep only the first-order ($a_1(\boldsymbol{x})$) term. Some work has explored the utility of higher-order terms, such as the quadratic term, but found it limited in extrapolative ability and physical justification (Herger et al., 2015).

Although pattern scaling implicitly attempts to approximate the Koopman operator - the perfect linear representation of the system - it is limited by its assumption of time-invariant, quasi-equilibrium dynamics. Truncating the operator with a finite dimensional approximation and using only a single predictive field (here, annual-mean temperature) further reduces its skill. Pattern scaling's inability to reproduce the pattern effect and other nonlinear/state-dependent feedbacks illustrates these limitations (Stevens et al., 2016; Giani et al., 2024). In Sect. 2.2.3, we explore alternative low-order approximations of the

Koopman operator to resolve these issues.

Pattern scaling could be extended to the Fokker-Planck operator by shifting and rescaling the full probability distribution based on global mean temperature, but this faces several limitations. Reliably estimating probability distributions requires large ensembles, which are computationally expensive. An alternate approach is to use long preindustrial control runs to generate the initial probability distribution and attempt to learn the linear scaling factor through the shorter SSP experiments. However,

a simple linear shift may not capture scenario-dependent changes in the shape of the distribution; recent emulation work with Gaussian process regression suggests these distributional shifts may be complex (Wang et al., 2025). When applying pattern scaling to the Fokker-Planck operator, we must also ensure the process does not violate the normalization of the distribution (i.e. the area under the curve must equal one).

We implement pattern scaling by calculating the global mean temperature anomaly and solving

$$\min_{a(\boldsymbol{x})} ||T(\boldsymbol{x}, t) - a(\boldsymbol{x})\overline{T}(\boldsymbol{x}, t)||^2. \tag{16}$$

Appendix A1 shows that pattern scaling has two irreducible sources of error: (1) an equilibrium term, where pattern scaling converges to the wrong steady-state value when forcing plateaus and (2) a memory term, where pattern scaling breaks down





when the system responds slowly compared to changes in the forcing. The latter cannot be accounted for within the pattern scaling framework, motivating the need for methods that explicitly capture memory.

### 2.2.2 Dynamical system/impulse response theory


Emulators that represent the climate system through response functions connect to fundamental principles of statistical mechanics and the Koopman/Fokker-Planck framework (Joos and Bruno, 1996; Hasselmann et al., 1997; Hasselmann, 2001; Lucarini et al., 2017; Orbe et al., 2018; Lembo et al., 2020; Fredriksen et al., 2021, 2023; Cimoli et al., 2023; Freese et al., 2024; Womack et al., 2025; Sandstad et al., 2025; Farley et al., 2025). Response function emulators relax the quasi-equilibrium

assumption, assuming instead that the current transient climate state is close to some baseline climate state that is in statistical equilibrium (generally preindustrial conditions). Perturbations to a field of interest are assumed to be small relative to magnitude of that field. These methods enable us to capture memory effects by integrating the entire forcing time history rather than only using the instantaneous forcing. One major benefit of this is that we can use them to represent regional shifts in surface warming patterns over time (the pattern effect) (Bloch-Johnson et al., 2024).

The use of different methods to derive response functions affects their utility as an emulator. A key assumption behind the Fluctuation Dissipation Theorem, for example, is that we have access to the governing equation, i.e. we are free to run large ensembles as needed. We begin this section assuming this is true, and relax this assumption later.

**Method II: The Fluctuation Dissipation Theorem.** In the case of a fully deterministic system with a zero initial condition, simply forcing our system with a spatially explicit unit impulse ($F(\boldsymbol{x}, t) = \delta(\boldsymbol{x}, t)$) is used to find the system's response function


$$T(\boldsymbol{x}, \boldsymbol{x}', t)|_{F(\boldsymbol{x}', t) = \delta(\boldsymbol{x}', t)} = R(\boldsymbol{x}, \boldsymbol{x}', t), \tag{17}$$

where perturbations are applied at each spatial location, $\boldsymbol{x}'$, to determine their influence on a location of interest, $\boldsymbol{x}$; pulses can also be applied at alternate times, $t'$, to determine how different time lags impact the response (e.g. seasonality), but we neglect these effects to simplify our analysis.

In this case, we can derive our response function directly without the need for an ensemble of simulations, but real systems are not this simple. Utilizing an impulse forcing naively in a chaotic system may lead to a single realization with behavior far from the expected forced response. For our nonlinear SDE, we use the Fluctuation Dissipation Theorem (FDT), to calculate a response function from an ensemble. Our system's response to a perturbation of magnitude $\varepsilon$ is given by

$$R(\boldsymbol{x}, \boldsymbol{x}', t) = \frac{\langle T_\varepsilon(\boldsymbol{x}, t) - T_0(\boldsymbol{x}, t)\rangle}{|\varepsilon(\boldsymbol{x}')|}, \tag{18}$$

where $T_0(\boldsymbol{x}, t)$ and $T_\varepsilon(\boldsymbol{x}, t)$ correspond to unperturbed and perturbed initial condition ensembles, respectively. More detail on this expression can be found in Marconi et al. (2008).

    With this definition, we implement the Fluctuation Dissipation Theorem by first spinning up a simulation to get a steady state distribution from which we draw an ensemble of initial conditions, $T_0(\boldsymbol{x}, t)$. We then create a copy of the initial condition ensemble with an additional small perturbation, $\varepsilon$, applied to each member, $T_\varepsilon(t)$, and simulate every member from both



ensembles for a scenario of interest. Applying Equation 18 then gives us the response function, which can use to emulate a variable of interest by convolving it with a forcing from a new scenario (Equation 12).

Both the stochastic and deterministic approaches only yield an accurate estimate of the true response function when the system is perturbed from a quasi-equilibrium rather than a transient state. For climate models, this is typically done with step change $CO_2$ experiments after a spin-up period. This method is common in the literature around climate response functions

and linear response theory (Lucarini et al., 2017; Lembo et al., 2020; Freese et al., 2024), though methods from the former two citations have not been applied to climate emulation and the latter does not reference formal response theory. Repeating this perturbation exercise at multiple background climate states can produce state-dependent response functions, but it is prohibitively expensive in practice.

Analogously to our discussion of using the Koopman vs. Fokker-Planck operator, there also exists an extension of the FDT

to probability distributions. This relationship is given by

$$R(\boldsymbol{x},\boldsymbol{x}',t) = -\langle T(\boldsymbol{x},t)s(T(\boldsymbol{x}',0))\rangle, \tag{19}$$

where $s(w) = \nabla \ln p(w)$ is the score function of the steady-state distribution and encodes how a small perturbation alters the system's dynamics; more details can be found in (Giorgini et al., 2024).

The score function captures the direction a distribution shifts in response to a perturbation, and correlating it with a climate

variable explains how the expectation of that variable shifts. Appendix A5 outlines the link between this approach and the Fokker-Planck operator. Analytical expressions for the score function are unavailable for most systems, necessitating machine learning techniques to learn the score function. This approach has achieved high skill in representing the response function for several systems (Giorgini et al., 2024), though it has not yet been applied to the full climate system. We do not explore it further in this work because of the machine learning infrastructure required to implement it.

The FDT faces accessibility issues in practice. First, there are high costs associated with this technique: a large ensemble of ESM runs is often prohibitively expensive. Second, there are also some configurations we simply cannot access: formal response theory assumes perturbations can be applied in a straightforward manner, which is not always the case. Because response functions are defined as a mapping from some perturbed input variable (e.g. $CO_2$ or radiative forcing) to an output variable of interest (e.g. temperature or precipitation), applying the FDT requires the ability to manually perturb a variable.

Climate models may not be configured to accommodate e.g. radiative forcing as an input. The FDT therefore cannot be applied to derive radiative forcing response functions, though this is possible through other methods (Womack et al., 2025).

**Method III: Deconvolution.** Without access to the true system to run specific perturbation experiments to find $R(\boldsymbol{x},\boldsymbol{x}',t)$, data-driven approaches can estimate it. Deconvolution has been used to calculate response functions in the climate emulation context to derive spatially explicit response functions mapping effective radiative forcing to temperature (Womack et al., 2025).

It implicitly approximates the Koopman operator by deriving response functions that nominally correspond to Equation 12. To derive the deconvolution algorithm, we assume the data we have (e.g. annual temperature anomaly) are taken from an ensemble average of a general scenario. We begin from the FDT (Equation 12), assuming that our experiment begins from a



quasi-equilibrium initial condition

$$T(\boldsymbol{x}, t) = \int\limits_{0}^{t} R(\boldsymbol{x}, s) F(t - s)\, ds. \tag{20}$$

Treating this expression discretely, we rewrite it as a matrix expression and invert to solve for $R(\boldsymbol{x}, t)$ from any general scenario

$$\mathbf{R} = \frac{\mathbf{F}^{-1}\mathbf{T}}{\Delta t}, \tag{21}$$

where $\mathbf{F}$ is a lower-triangular matrix with $F_{t=0}$ along the diagonal, $F_{t=1}$ on the first off-diagonal, and so on (a Toeplitz matrix), and $\mathbf{T}$ is a matrix of temperature values with rows corresponding to the time dimension and columns corresponding

to the spatial dimension. A more in-depth exploration of this process can be found in Womack et al. (2025). As written here, deconvolution aggregates spatial interactions (i.e. does not include an $\boldsymbol{x}'$ term), cutting down on data requirements. Extensions of this procedure can account for spatial interactions, though they require additional experiments with varying spatial forcings.

    In practice, noisy data require us to apply regularization to Equation 21 to ensure matrix stability. We instead solve

$$\min_{\mathbf{R}} ||\mathbf{RF} - \mathbf{T}||^2 + \alpha ||\mathbf{R}||^2, \tag{22}$$

where $\alpha$ is the hyperparameter denoting the strength of our ridge regression. This simple ridge regression is equivalent to placing a Gaussian prior on the response function and assuming that the simulated temperature data we collect are corrupted by Gaussian noise. We discuss the rationale of Gaussian noise further in Appendix B and outline our approach to tune the hyperparameter $\alpha$ through *maximum a posteriori* optimization.

    Deconvolution can be applied to any general scenario that begins from a quasi-equilibrium initial condition. However, since

we require an explicit matrix inverse to perform deconvolution, it is sensitive to the frequency spectrum of the forcing data. If the eigenvalues of the matrix $\mathbf{F}$ are very small (corresponding to near-zero frequencies) or the system is very noisy (corresponding to large differences in magnitudes between frequencies), the matrix becomes ill-conditioned, leading to an unstable response function. To illustrate these challenges, an explicit frequency-based derivation is included in Appendix A2. In practice, we regularize the system to avoid these issues (see Appendix B for details).

**Method IV: Modal Fitting.** Modal fitting is another data-driven technique to calculate response functions that retains some physical interpretability by explicitly representing the climate's response to a forcing as a series of decaying exponentials. The decay rates then represent the various timescales of the climate system (e.g. shallow vs. deep ocean heat uptake) and the modes represent how those timescales interact spatially. It has been used for tasks such as estimating effective radiative forcing and recently for climate emulation (Fredriksen et al., 2021, 2023; Sandstad et al., 2025).

To connect this approach to our framework, we begin from the same set of assumptions as deconvolution, but make the additional assumption that our response function is exactly a decaying exponential; in this case, our response function is exactly a Green's function as described in Appendix A3. We start from a restatement of Koopman response function definition (Equation 12)

$$G(\boldsymbol{x}, \boldsymbol{x}', t) = e^{\mathcal{L}(\boldsymbol{x}, \boldsymbol{x}')t}, \tag{23}$$



where $\boldsymbol{x}$ and $\boldsymbol{x}'$ track spatial interactions as before. We assume we can represent the Koopman operator with a finite, linear operator, $\mathcal{L}$ (Appendix A4).

We then diagonalize the matrix $\mathcal{L}$ though an eigenvalue decomposition, giving

$$G(\boldsymbol{x},\boldsymbol{x}',t) = e^{v(\boldsymbol{x},n)\Lambda(n,n)v^{-1}(n,\boldsymbol{x}')t}, \tag{24}$$

where $\Lambda(n,n)$ and $v(\boldsymbol{x},n)$ are matrices containing the system's eigenvalues and eigenvectors, respectively, and $n$ is the mode

number. Since the matrix exponential respects similarity transformations, we rewrite this exactly as the summation

$$G(\boldsymbol{x},\boldsymbol{x}',t) = \sum_{i=1}^{k} v(\boldsymbol{x},n_i)e^{\lambda_i t}v^{-1}(n_i,\boldsymbol{x}'), \tag{25}$$

where $k$ is equal to total the number of eigenvalues in the system. In the case of a climate model, the dimension of $k$ is equivalent to the number of spatial dimensions. This may be much higher than the true number of modes that are significant in determining e.g. the temperature response of the system. Instead of the explicit form above, we typically see an alternate

implementation, such as that in (Fredriksen et al., 2021, 2023) and (Sandstad et al., 2025). These show that one can fit an alternate form given simply by

$$G(t) \approx R(t) = \sum_{i=1}^{3} \alpha_i e^{\lambda_j t}, \tag{26}$$

where using just three timescales (inter-annual, inter-decadal, and inter-centennial) is sufficient to represent the global mean behavior of the climate system; these methods specify a range/initial guess of timescales to initialize the optimization routine.

As we are implementing this at a grid cell level, we opt for a hybrid approach, given by

$$R_i(t) = \sum_{j=1}^{3} \alpha_{i,j} e^{\lambda_i t}, \tag{27}$$

where $i$ indicates the grid cell/region of interest, and $j$ denotes the contribution from each timescale in a given region. We use the three timescales given above as the initial guess for each lambda, along with an initial guess for $\alpha_{i,j} \forall i = j$, assuming that one mode is dominant for each box.

We thus need to solve

$$\tilde{T}(\alpha_{i,j},\lambda_i) = \int_{-\infty}^{t} R_i(s)F(t-s)\,ds, \tag{28}$$

$$\min_{\alpha_{i,j},\lambda_i} ||T - \tilde{T}(\alpha_{i,j},\lambda_i)||^2. \tag{29}$$

For climate applications, the decay rates ($\lambda_i$) can span several orders of magnitude, which are difficult for the optimizer to identify, even with normalization. This is exacerbated by the need to solve for the eigenvectors simultaneously, which are also

likely to have values that span several orders of magnitude; using more sophisticated optimization techniques than we apply in our test case could potentially resolve this issue. When implementing this algorithm, we follow Fredriksen et al. (2021), providing an initial guess of the correct order of magnitude to our optimizer.



Modal fitting has two major benefits. First, by truncating the leading modes, we reduce the dimensionality of the problem without the need for e.g. Empirical Orthogonal Functions (EOFs) or a Singular Value Decomposition (SVD). Second, we require all $\Re(\lambda_i) < 0$ (the real component of $\lambda_i$) to ensure response functions to decay to zero as $t \to \infty$, a requirement not imposed on e.g. deconvolution and DMD. Because it is a best-fit problem, it naturally damps noise, making it well suited to systems with strong internal variability. However, this method can also be sensitive to local minima, requiring multiple iterations or a stochastic fitting procedure to alleviate this issue. Fitting may also be expensive on fine grids, since the number of eigenpairs scales with grid size, though we may not require all eigenpairs to accurately emulate the system.

### 2.2.3 Operator-based emulation

The most general class of emulators are those that aim to directly approximate the Koopman operator. Every previous emulator can be thought of as a specific case of this general operator framework. Tebaldi et al. (2025) do not include operator-based emulators in their classification, as they are not typically referred to explicitly as emulators. However, we classify them as such to facilitate communication across disciplines with similar prediction goals.

The most common data-driven approximations of the Koopman operator are Dynamic Mode Decomposition (DMD) and Extended DMD (EDMD) (Schmid, 2010; Williams et al., 2015). Schmid (2010) developed DMD to extract dynamic information from fluid flows, and it has since been used to identify dominant modes of variability within the climate system, including El Nino–Southern Oscillation, North Atlantic Oscillation, and Pacific Decadal Oscillation (Kutz et al., 2016; Gottwald and Gugole, 2020; Navarra et al., 2021; Franzke et al., 2022; Navarra et al., 2024; Mankovich et al., 2025). Under specific conditions, DMD provides a finite-dimensional approximation of the Koopman operator (Schmid, 2022). EDMD expands this idea to approximate Koopman eigenvalues and eigenfunctions directly (Williams et al., 2015). The bulk of the work surrounding EDMD is theoretical (Haseli and Cortés, 2019; Netto et al., 2021), as in practice it has several limitations that we outline later in this section.

**Method V: Dynamic Mode Decomposition (DMD).** DMD assumes that the climate response is linear in $w$ with respect to an operator. If this is the true Koopman operator, this assumption holds by definition, provided it acts on the entire infinite space of statistical climate fields, $g(w)$. In practice, this leads to limitations based on how accurate the assumption of linearity is, which depends on the choice of variables; this approximation may hold better for a variable such as temperature, rather than precipitation. To derive DMD, we begin from Equation 9 applied to our variable of interest

$$\frac{\partial}{\partial t} T(\boldsymbol{x}, t) = \mathcal{K}(\boldsymbol{x}, \boldsymbol{x}') T(\boldsymbol{x}', t) + F(\boldsymbol{x}, t). \tag{30}$$

DMD assumes that we separate our data in discrete snapshots, $T(\boldsymbol{x}, t_0), T(\boldsymbol{x}, t_1), \ldots, T(\boldsymbol{x}, t_n)$, which we assume are linearly related

$$T_{n+1} = \mathcal{L} T_n + F_n, \tag{31}$$

where we have used the subscript $n$ as shorthand for $t_n$ and omitted the spatial dimension for conciseness. By discretizing, we are no longer solving for the exact Koopman operator (as in the previous case), which we now denote $\mathcal{L}$. This notation



is standard in DMD literature. The traditional DMD algorithm assumes autonomous dynamics, omitting the forcing term. Equation 31 is referred to as DMD with control (DMDc) (Proctor et al., 2016), and has only recently been studied in the climate context (Mankovich et al., 2025).

  To implement DMD, we collect our snapshots into matrices and invert this system, solving for $\mathcal{L}$

$$\mathcal{L} = [\mathbf{T}_{n+1} - \mathbf{F}_n]\,\mathbf{T}_n^+, \tag{32}$$

where the superscript $+$ denotes the Moore-Penrose pseudo-inverse of a matrix (required as it unlikely $x$ and $t$ will be the same dimension, i.e. it is unlikely $\mathbf{T}$ is a square matrix) and $\mathbf{F}$ denotes a forcing matrix with the same dimension as our data; assuming well-mixed forcing means each row is identical in the forcing matrix. This is the simplest form of DMD, though in practice the Singular Value Decomposition (SVD) is often used to further reduce the dimensionality of the problem. This also increases the algorithm's robustness relative to real-world systems that are subject to noise (Schmid, 2010).

This approach suffers mainly from its strong assumption of linear dynamics, which can break down for complex systems. Its success in identifying the dominant modes of variability in the climate suggests it may have utility as an explicit emulation technique (Kutz et al., 2016; Gottwald and Gugole, 2020; Franzke et al., 2022); future work will apply DMD to a full scale climate model to test this hypothesis. Unfortunately, DMD only provides a reliable estimate for the Koopman operator if it acts on a large set of statistical fields (more than simply the temperature anomaly when considering the full climate system) 

and/or the dynamics governing the evolution of that quantity (or quantities) is linear, which is not the case in general. While the dynamics producing the base climate state are nonlinear, the success of methods such as pattern scaling suggest the dynamics of anomalies may be close to linear. DMD assumes all hidden variables are accounted for and the observed quantities fully describe the (linear) dynamics of our anomaly of interest. For example, the atmospheric temperature may be significantly influenced by heat uptake in the deep ocean, which, if it is not explicitly accounted for, will lead to errors when applying 

DMD. This motivates the need for a better algorithm for approximating the Koopman operator.

  **Method VI: Extended DMD (EDMD).** As the baseline DMD algorithm is only able to approximate the Koopman operator in specific contexts, EDMD instead frames the problem such that we are deliberately trying to approximate the eigenvalues and eigenfunctions of the Koopman operator. This, ideally, leads to more reliable approximation than DMD and thus, a better emulator.

EDMD was introduced by Williams et al. (2015) as an explicit attempt to approximate the Koopman operator. The EDMD procedure involves projecting variables of interest into a higher dimensional space that has a richer representation of the system dynamics. As an example, we consider the problem of emulating precipitation anomaly using global mean temperature anomaly as the forcing. Precipitation may depend on the global mean temperature, $\overline{T}(t)$, but it also may depend on higher order or nonlinear terms, such as $(\overline{T}(t))^2$, $\cos(\overline{T}(t))$, $\tanh(\overline{T}(t))$, etc. To implement EDMD, the user must select a set of 

basis functions, $\phi(\cdot)$, such as these, that provide a better representation of the system dynamics than in the purely linear DMD case. Typical choices of basis functions as described by the original EDMD manuscript are Hermite polynomials, radial basis functions, and discontinuous spectral elements (Williams et al., 2015).





After choosing a set of basis functions, the EDMD problem statement is exactly the same as the original DMD algorithm. Solve for $\tilde{\mathcal{K}}$ from

$$\phi(T_{n+1}) = \tilde{\mathcal{K}}\phi(T_n) + \psi(F_n), \tag{33}$$

where $\psi(\cdot)$ is the basis chosen for the forcing, and can be the same or different than the forcing for the quantity of interest. We use $\tilde{\mathcal{K}}$ here as we are explicitly trying to approximate the Koopman operator. We ensure the basis includes the physical field of interest, e.g. $\phi(\mathbf{T}) = \left[\mathbf{T}, \mathbf{T}^2, \mathbf{T}^3, \dots\right]$, where the first entry is the physical field. As in the case with DMD, we solve this as

$$\tilde{\mathcal{K}} = \left[\phi(\mathbf{T}_{n+1}) - \psi(\mathbf{F}_n)\right]\phi^+(\mathbf{T}_n), \tag{34}$$

which we can use an SVD to solve more efficiently and reduce the influence of noise on the system. When applying this method, we first use Equation 33 with an appropriate initial condition to emulate the solution in our high-order basis. We then must project our solution back into physical space. Since we chose our basis to include the original physical coordinate, this is done by truncating the emulator output and keeping only the entries corresponding to $\mathbf{T}$.

This method has seldom been applied to climate problems (Navarra et al., 2024), likely due to the limitations acknowledged in Navarra et al. (2021), particularly the dimensionality of the problem. For a full climate model, DMD requires a matrix solve of dimension $(N_{\text{lat}} \times N_{\text{lon}})^2$ for a single variable, which is extremely costly. In the case of EDMD, this dimension grows with every basis function used. To accurately represent the Koopman operator for the climate system, we potentially require many more variables and many basis functions, causing the problem to rapidly increase in complexity, though this may be alleviated by emulating EOFs rather than gridded data. As with DMD, EDMD implicitly assumes no hidden variables, though the choice of basis function can help alleviate this issue; e.g. if the hidden variables are higher order terms, EDMD may be able to represent them accurately. The selection of basis functions typically requires some experimentation though, as it can be difficult to predict which set of functions will be best suited for a given application; exploiting physical relationships such as the logarithmic relationship between $CO_2$ concentration and temperature may help alleviate this issue, however. More work is required to fully characterize the utility of EDMD for the climate system.

## 2.3 Experimental overview

We conduct four experiments to evaluate these emulation techniques across a range of potential high-error scenarios. We outline a climate box model with a simple local energy balance ODE in Sect. 2.3.1 and Sect. 2.3.2, followed by a nonlinear, cubic Lorenz system in Sect. 2.3.3. Experiments using these two models highlight the following potential sources of error: (1) memory effects, Fig. 1 (a); (2) hidden variables, Fig. 1 (b); (3) noise, Fig. 1 (c); (4) weak nonlinearities, Fig. 1 (d). We then describe forcing scenarios applied to each system in Sect. 2.3.4.

### 2.3.1 Experiments 1 and 2: Climate Box Model

A classical box model is a standard, easily-interpretable model for temperature evolution. We use this idealized box model as it is the simplest system that includes the pattern effect and it is not necessarily meant to replicate CMIP experiments. We assume





the form of this model is given by a simple local energy balance

$$C(\boldsymbol{x})\frac{\partial T(\boldsymbol{x},t)}{\partial t} = \lambda(\boldsymbol{x})T(\boldsymbol{x},t) + R(\boldsymbol{x},t) + \nabla \cdot \mathbf{F}(\boldsymbol{x},t), \tag{35}$$


similar to Armour et al. (2013) and Giani et al. (2024). $C(\boldsymbol{x})$ is the local effective heat capacity, $T(\boldsymbol{x},t)$ is the local temperature anomaly, $\lambda(\boldsymbol{x})$ is the local feedback parameter, $R(\boldsymbol{x},t)$ is the forcing function, and $\nabla \cdot \mathbf{F}(\boldsymbol{x},t)$ is the anomaly in heat flux divergence; parameters for this model are listed in Table 2. Furthermore, we assume that the forcing function can be linearly decomposed as a constant-amplitude spatial pattern and a variable time series: $R(\boldsymbol{x},t) = r(\boldsymbol{x})R(t)$.

We consider two configurations for our box model. The first corresponds to a horizontally coupled three box system representing atmospheric boxes over land, low-latitude ocean, and high-latitude ocean; $\nabla \cdot \mathbf{F}(\boldsymbol{x},t) = -k(\boldsymbol{x})\nabla T(\boldsymbol{x},t)$. We assume a constant diffusivity and discretize as, $\nabla \cdot \mathbf{F}(\boldsymbol{x},t) = -k(T_{i+1}(t) - T_i(t))$, where $i$ refers to the index of each box. We assume uniform forcing into each box, and use this configuration for experiments one and three (memory effects and noise; noise details can be found in Sect. 2.3.2). The second configuration corresponds to a vertically coupled two box system representing

the atmosphere and the ocean; this has the same form as the previous case, with the caveat that there is no forcing applied into the oceanic box. We use this configuration for experiment two (hidden variables). We begin this system from a zero initial condition, aiming to simulate the temperature anomaly, rather than the absolute temperature.

**Table 2.** Parameters for the three box model, adapted from Giani et al. (2024). The heat capacity of each box is given in terms of the effective water depth, $h(\boldsymbol{x})$: $C(\boldsymbol{x}) = \rho_w c_w h(\boldsymbol{x})$, where $\rho_w$ and $c_w$ are the density and specific heat capacity of water, respectively. *Land*, *Low*, and *High* refer to atmospheric boxes over land, low-latitude ocean, and high-latitude ocean, respectively.

| Parameter | Symbol | *Land* | *Low* | *High* |
|---|---|---|---|---|
| Effective Water Depth (m) | $h(\boldsymbol{x})$ | 5 | 150 | 1500 |
| Local Feedback (Wm$^{-2}$K$^{-1}$) | $\lambda(\boldsymbol{x})$ | -0.86 | -2.0 | -0.67 |

#### 2.3.2   Experiment 3: Noisy Box Model

As the default configuration for our box model is purely deterministic, we add a stochastic noise term to the forcing to replicate

the impact of inter-annual variability on the real climate system. To ensure the impact of this variability is similar to that of the true system, we use CMIP6 *piControl* experiments to estimate the magnitude of the variability. Namely, we compute the standard deviations of piControl runs for three climate models (ACCESS-ESM1-5, MIROC6, MPI-ESM2-LR) and set the magnitude of the variability as the multi-model average $\sigma = 0.117$K (Dix et al., 2023; Tatebe and Watanabe, 2023; Wieners et al., 2023).

#### 2.3.3   Experiment 4: Cubic Lorenz System

As the previous experiments are all defined by an operator which is linear in the quantity of interest, we additionally implement a weakly nonlinear, cubic Lorenz system. This provides a representation of the atmosphere that includes chaos, allowing us to



test the limits of these emulation techniques. In the standard Lorenz equations that represent a simplified model of atmospheric convection (Lorenz, 1963), the steady state is a linear function of $\rho$, and the mean heat flux ($\langle XY \rangle = \langle Z \rangle$) is very nearly linear (Souza and Doering, 2015). We modify the system to the cubic form shown below to illustrate another failure mode of simple pattern scaling: the quasi-equilibrium value may not be a linear function of the forcing.

The cubic Lorenz equations are defined by the system

$$\frac{\partial}{\partial t}X = \sigma(Y - X), \tag{36}$$

$$\frac{\partial}{\partial t}Y = -(Z + \alpha Z^3)X + \rho(t)X - Y, \tag{37}$$

$$\frac{\partial}{\partial t}Z = XY - \beta Z, \tag{38}$$

with $\alpha = 1/1000$. The steady-state mean of both $X$ and $Y$ are zero, while the steady-state behavior of $\langle Z \rangle$ is determined by $\rho(t)$. Values for $\rho(t)$ are chosen such that nonlinearities are weak, as all linear methods are expected to break down in the presence of strong nonlinearities. These vary between experiments and are outlined in Table 4. We initialize this system through an initial condition ensemble starting from $\rho(t) = 28$ with white noise applied to perturb the starting positions of each ensemble member.

### 2.3.4 Scenarios

We consider four scenarios of interest for both the box model and cubic Lorenz system, focusing on scenarios which have CMIP analogues: (1) *Abrupt*, an abrupt increase in forcing, (2) *High Emissions*, an exponential increase in forcing, (3) *Plateau*, an exponentially increasing in forcing that levels off, and (4) *Overshoot*, a forcing that sharply increases and decreases. Descriptions of each scenario are given in Table 3. Figure 3 shows ODE-integrated solutions for each scenario in each experiment, and descriptions of experimental parameters can be found in Tables 4 and 5.

### 2.4 Evaluation

To evaluate each emulation technique, we utilize Normalized Root Mean Square Error (NRMSE, Equation 39) given as a percentage, as our primary evaluation metric:

$$\text{NRMSE} = \frac{100}{\overline{g}(w_k)}\sqrt{\frac{\sum_{k=1}^{N_{\text{years}}}\left(g(w_k) - \hat{g}(w_k)\right)^2}{N_{\text{years}}}}. \tag{39}$$

$\overline{g}(w_k)$ indicates the mean of our quantity of interest over the period error is calculated over. We calculate NRMSE with respect to the entire time series. To compare performance across training datasets, we train each emulator on one scenario at a time, testing against the others which are held out from the training (e.g. train on *Abrupt* and test on *High Emissions*).

We implement an alternate protocol for the cubic Lorenz system as there is no ground-truth to compare with due to chaos. Instead, we compare the skill of each emulator when training on only a subset of the ensemble members for that experiment. For example, given $n_{\text{ensemble}}$ ensemble members for a given experiment, we construct a subset of $n$ ensemble members without



**Table 3.** Conceptual overview of forcing scenarios considered in this work. These scenarios are used in all experiments outlined in Sect. 2.3, and lists of experiment-specific parameters for each scenario can be found in Tables 4 and 5.

| Scenario | Short Description |
|---|---|
| *Abrupt* | An abrupt doubling of $CO_2$ concentration; corresponds roughly to the *Abrupt2xCO2* CMIP experiment. |
| *High Emissions* | An exponential increase of $CO_2$ concentration in time; corresponds roughly to *SSP585*. |
| *Plateau* | An increase in $CO_2$ concentration in time that follows a hyperbolic tangent, increasing exponentially and then tapering off; corresponds roughly to *SSP245*. |
| *Overshoot* | An increase in $CO_2$ concentration in time that follows a Gaussian profile, increasing and decreasingly rapidly; inspired by *SSP119*, but decreases more quickly. |

**Table 4.** Forcing scenarios for each experiment, with the upper half of each row corresponding the box model and the lower half of each row corresponding to the cubic Lorenz system. Parameters for the box model experiments are based on Giani et al. (2024) and (Armour et al., 2013) and parameters for the cubic Lorenz system are chosen such that the system exhibits weakly nonlinear behavior. $H(t)$ is the Heaviside step function, and parameters for these scenarios are listed in Table 5.

| Scenario | Functional Form |
|---|---|
| *Abrupt* | $F(t) = F_{abr} H(t)$ |
| | $\rho(t) = \rho_{0,abr} + \rho_{1,abr} \tanh(t - \eta_{abr})$ |
| *High Emissions* | $F(t) = F_{high} \exp(t/\tau_{high})$ |
| | $\rho(t) = \rho_{0,high} + \rho_{1,high} \exp(t/\eta_{high})$ |
| *Plateau* | $F(t) = F_{plat} + F_{plat} \tanh(\omega_{plat}(t - \tau_{plat}))$ |
| | $\rho(t) = \rho_{0,plat} + \rho_{1,plat} \tanh(\omega_{plat}(t - \tau_{plat}))$ |
| *Overshoot* | $F(t) = F_{over} \exp(-(t - \tau_{over})^2/(2\sigma^2))$ |
| | $\rho(t) = \rho_{0,over} + \rho_{1,over} \exp(-(t - \eta_{over})^2/(2\sigma^2))$ |

replacement, where $n = 1 : n_{\text{ensemble}} - 1$, and train our emulator from that subset. We then test the emulator's skill in emulating the mean response given the ensemble average forcing. We repeat this subsampling exercise 10 times, recording the average performance over those trials. For the noisy three box model, we use the same protocol, additionally presenting the ground 585 truth of emulating the noiseless three box model.





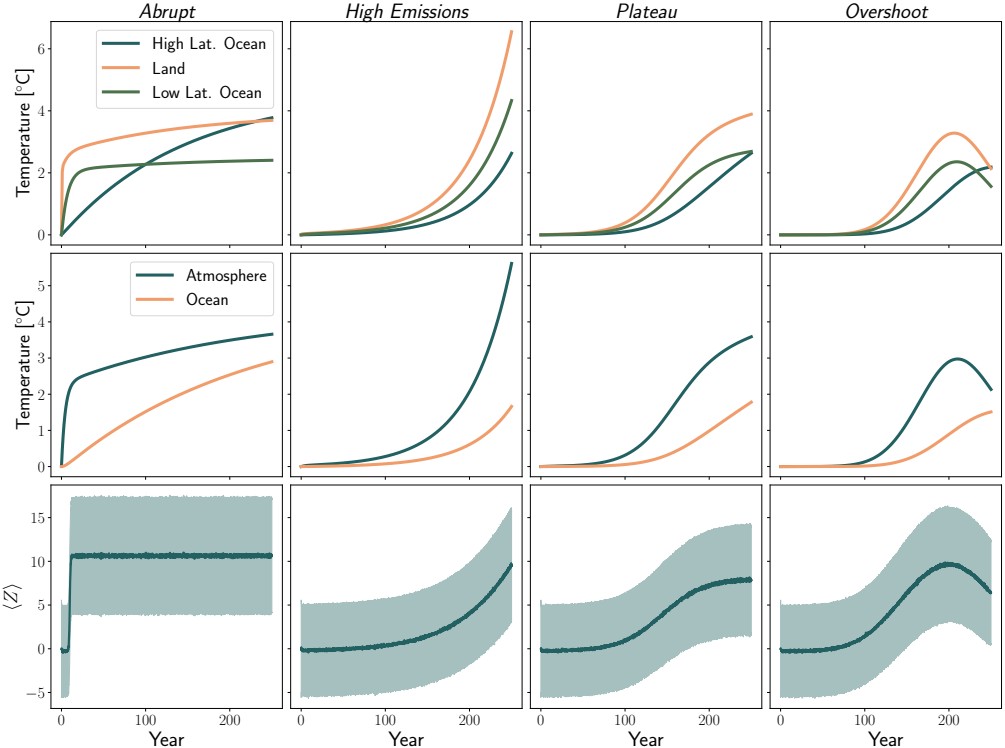

**Figure 3.** ODE-integrated solutions for the three box model (top), two box model (middle), and cubic Lorenz system (bottom) for the (from left to right) *Abrupt*, *High Emissions*, *Plateau*, and *Overshoot* scenarios. $D = 0.55$ [Wm$^{-2}$K$^{-1}$] for the three box experiment and $D = 0.7$ [Wm$^{-2}$K$^{-1}$] for the two box experiment. For the cubic Lorenz problem we show the mean value of $Z$ over 5,000 ensemble members as a line, and the shaded region indicates its standard deviation. Values shown are anomalies relative to a baseline of $T = 0$ (experiments one through three) or $\rho = 28$ (experiment four).

## 3 Results

Section 3.1 presents a summary of results across each of the emulation techniques outlined in Sect. 2.2 when emulating the simplified climate systems presented in Sect. 2.3, with subsequent sections highlighting key results from individual experiments. Section 3.2 contains the results for the three box model with significant memory effects (Fig. 1 (a)); the three boxes represent atmospheric boxes over the land, low-latitude ocean and high-latitude ocean. We then report emulator performance on the restricted two box model in Sect. 3.3. In this case we highlight the issue of hidden variables (Fig. 1 (b)) by only giving the emulators access to the temperature anomaly in only one of the two boxes during training; the two boxes represent an atmospheric and oceanic box (forcing only into the atmosphere). This is followed by a version of the three box model with a stochastic forcing to test the robustness of each method to noise (Fig. 1 (c)). Finally, we showcase results for the nonlinear, cubic Lorenz system in Sect. 3.5 (Fig. 1 (d)), which tests emulator performance in the presence of chaos and weak nonlinearities.



**Table 5.** Scenario parameters used for the experiments in this study. Values for $\rho_0$ are listed in the order *Abrupt*, *High Emissions*, *Plateau*, and *Overshoot*. Box-model parameters have physical units to output temperature; the cubic-Lorenz parameters are dimensionless.

| Box Model | | Cubic Lorenz System | |
|---|---|---|---|
| Parameter | Value | Parameter | Value |
| - | - | $\rho_0$ | $[45, 28, 40, 28]$ |
| $F_{abr}$ | $3.7 \, \mathrm{W\,m^{-2}}$ | $\eta_{abr}$ | 10 |
| | | $\rho_{1,abr}$ | 17 |
| $F_{high}$ | $\dfrac{8.5 \, \mathrm{W\,m^{-2}}}{\exp(\tau_f/\tau_{high})}$ | $\rho_{1,high}$ | $\dfrac{30}{\exp(\eta_f/\eta_{high})}$ |
| $\tau_f$ | $250 \, \mathrm{yr}$ | $\eta_f$ | 250 |
| $\tau_{high}$ | $50 \, \mathrm{yr}$ | $\eta_{high}$ | 50 |
| $F_{0,plat}$ | $2.25 \, \mathrm{W\,m^{-2}}$ | $\rho_{1,plat}$ | $\dfrac{12}{\tanh(5)}$ |
| $F_{1,plat}$ | $\dfrac{2.25 \, \mathrm{W\,m^{-2}}}{\tanh(\omega_{plat}\tau_{plat})}$ | $\tau_{plat}$ | 150 |
| $\omega_{plat}$ | $1/50 \, \mathrm{yr^{-1}}$ | $\omega_{plat}$ | $1/50$ |
| $F_{over}$ | $4 \, \mathrm{W\,m^{-2}}$ | $\rho_{1,over}$ | 30 |
| $\tau_{over}$ | $200 \, \mathrm{yr}$ | $\eta_{over}$ | 200 |
| $\sigma_{over}$ | 42.47 | $\sigma$ | 50 |

## 3.1 Overall emulator performance

Figure 4 summarizes emulator performance in terms of Normalized Root Mean Square Error (NRMSE) across all four experiments. For each experiment, there are four possible train/test scenarios (*Abrupt*, *High Emissions*, *Plateau*, and *Overshoot*). We test on one scenario and train against the remaining three, showing median NRMSE over all train/test combinations. For experiments two and four, the pattern scaling emulator is trained to map forcing to quantity of interest, as these experiments do not have a global mean temperature equivalent. Results for deconvolution are shown using the regularization presented in Appendix B. Error values are calculated with a constant 40 ensemble members for experiment three and 4,000 ensemble members for experiment four.

Response function based emulators (the FDT, deconvolution, and modal fitting methods) generally outperform other approaches, demonstrating consistently lower NRMSE across most experiments. The FDT is particularly reliable relative to all other methods, yielding consistently low errors across all four test cases, indicating its robustness regardless of scenario; while it has higher error in the cubic Lorenz case, this is primarily a function of ensemble size (see Sect. 3.5). As FDT response functions are, in principle, equation-driven rather than data-driven, they provide the perfect solution given a linear system (experiments one through three) or enough realizations (experiment four). Deconvolution similarly performs well across all experiments, while modal fitting has high performance in experiments one, two, and three; both of these methods exhibit higher errors in experiment four. For deconvolution, this is due to its sensitivity to noise as discussed in Sect. 2.2.2, while modal fit-





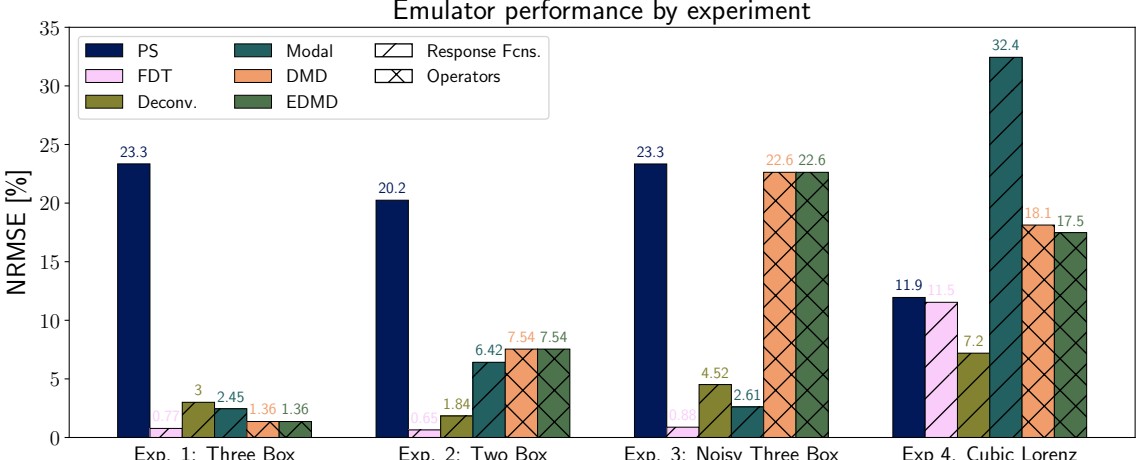

**Figure 4.** Summary of emulator performance over all experiments considered in this work. For each experiment, there are four scenarios. We show the median NRMSE value across all scenario train and test combinations, excluding the trivial case of training and testing on the same dataset. Error values are calculated with 40 ensemble members for experiment three and 4,000 ensemble members for experiment four. Emulator abbreviations are as follows: PS - Pattern Scaling, FDT - Fluctuation Dissipation Theorem, Deconv. - Deconvolution, Modal - Modal Fitting, DMD - Dynamic Mode Decomposition, EDMD - Extended DMD. Diagonal hatching indicates response function emulators, while cross hatching indicates operator-based emulators.

ting suffers because of an inability to reliably separate timescales and the need for an accurate initialization for its unknown parameters, which we discuss in Sect. 3.4.

In contrast, pattern scaling consistently underperforms, exhibiting the highest error in all experiments except for the cubic Lorenz case. This is most likely due to the presence of strong memory effects in the box models, which pattern scaling cannot capture by definition. DMD and EDMD outperform pattern scaling in experiments one and two, but exhibit much more variable performance in experiments three and four. For the first three experiments, DMD and EDMD produce identical results. This is because the models in these experiments are purely linear, and the use of any higher-order basis for EDMD leads to a drop in skill. These methods struggle with the noisy three box model, and more in-depth results can be found in Sect. 3.4. While theory 620 suggests DMD/EDMD would not be well-suited for the restricted two box problem due to the presence of hidden variables, they outperform pattern scaling in practice. This is likely due to the simplicity of the problem, and more complex dependencies on hidden variables would likely lead to further decreases in skill. The main advantage of EDMD over DMD begins to become apparent in the cubic Lorenz experiment, where moving to a 3rd Hermite polynomial basis allows it to slightly outperform its linear counterpart, though the variability in the system (Fig. 3) is a greater magnitude than this improvement in skill.

**3.2 Experiment 1: Three Box Model**

The three box model experiment is meant to benchmark the baseline performance of each technique in the presence of strong memory effects (Fig. 1 (a)). Figure 5 summarizes the results of four emulation techniques (pattern scaling, deconvolution,

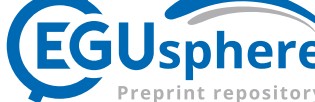

modal fitting, and DMD) when trained and tested on different scenario combinations, while Fig. 6 compares the true (ODE-integrated) solution to that obtained using the Fluctuation Dissipation Theorem.

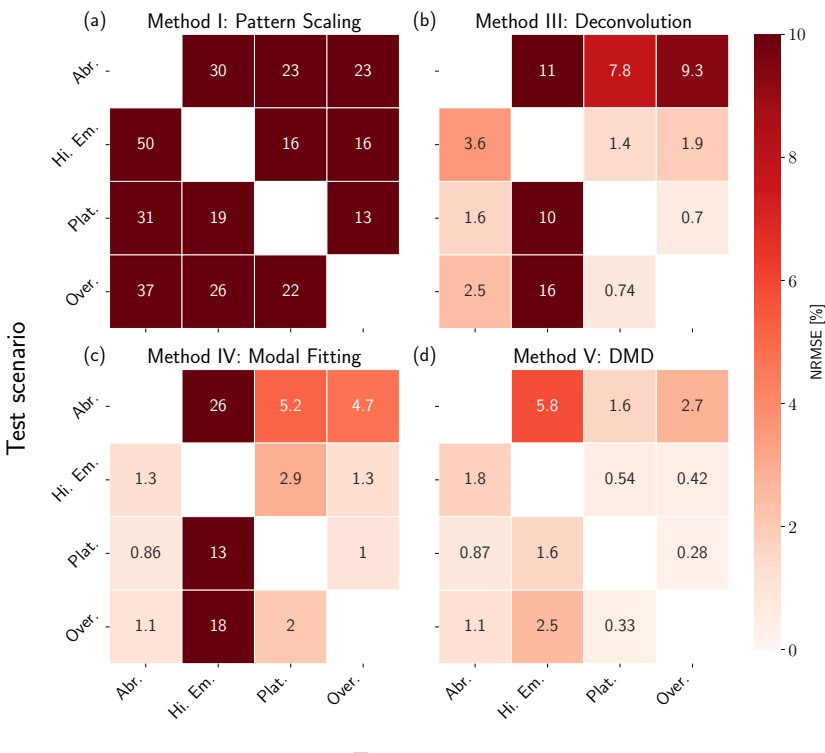

**Figure 5.** NRMSE heatmaps for pattern scaling (a), deconvolution (b), modal fitting (c), and DMD (d) emulators trained and tested against the three box model. Results are shown in percentages, where lighter values correspond to lower error (higher performance) and darker values correspond to higher error (lower performance). Scenarios used for training are shown on the x-axis, while scenarios used for testing are shown on the y-axis. We do not include results for training and testing on the same dataset.

Pattern scaling (Method I) consistently underperforms relative to the other techniques presented in this section, exhibiting the highest NRMSE values for all train/test combinations. It fails across almost every scenario due to the influence of long timescales on the global mean temperature (strong memory effects). This experiment highlights pattern scaling's brittleness when key assumptions, such as exponential forcing (Giani et al., 2024), are violated. These assumptions are consistent in most ScenarioMIP experiments however, leading to higher performance in practice relative to this simple example (Wells et al.,

635    2023).

Applying deconvolution (Method III) leads to much higher performance than pattern scaling when trained on either *Abrupt*, *Plateau*, or *Overshoot*, but sees a drop in performance when trained on *High Emissions*. This is because the true solution is an eigenfunction of the forcing (i.e. both the temperature response and forcing are exponentials), so the system is effectively





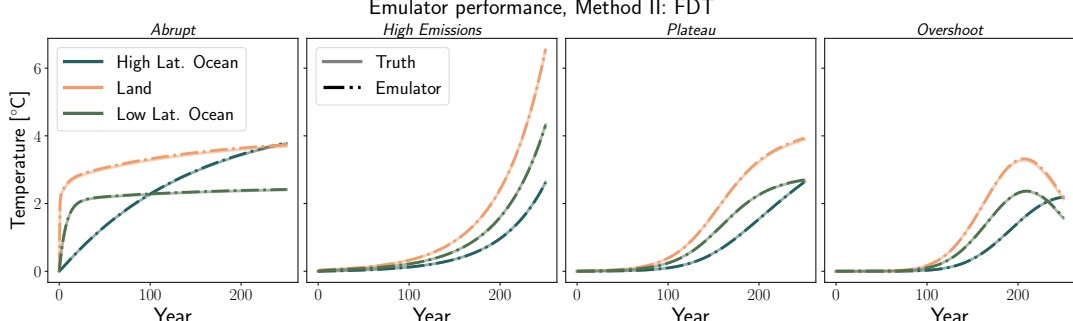

**Figure 6.** Fluctuation Dissipation Theorem emulator performance for three box model scenarios. Solid line shows ground truth (ODE-integrated) solution, while dot-dash shows emulated solution.

characterized by a single timescale, that of the forcing. Deconvolution loses skill due to difficulties identifying all the timescales
in the system, leading to extrapolation errors when training on this scenario. When trained on either *Plateau* or *Overshoot*, we see errors in emulating *Abrupt*, meaning that the emulator has not learned the true system response despite relatively high performance in emulating the other scenarios. This is due to ill-conditioning of the **F** matrix in these scenarios, leading to a response function that overfits these data; we discuss the limitations of training deconvolution with these scenarios further in Sect. 4.

Modal fitting (Method IV) exhibits two interesting properties: (1) training on *High Emissions* leads to poor extrapolative capability and (2) training on *Abrupt* leads to the highest performance overall. The first is also caused by the solution being an eigenfunction of the forcing. It is difficult for the optimization routine to determine the correct timescales, even when initialized near the true values. This is true to a lesser degree in *Plateau* and *Overshoot*, which also do not display clean separation of time scales like *Abrupt*.

DMD (Method V) is able to capture all relevant timescales and interactions regardless of the scenario, with a maximum of 5.8% NRMSE across all train/test combinations; this level of error results from training on *High Emissions* and testing on *Abrupt*, as was the case with the modal fitting emulator. The method's high skill here is due to the governing dynamics being purely linear and there being no hidden variables, meaning all assumptions for applying DMD are accurate. Results for EDMD (Method VI) are omitted from this section as they are identical to DMD.

The Fluctuation Dissipation Theorem (Method II) has consistently high performance across all scenarios considered, with NRMSE values of 0.80%, 0.50%, 0.75%, and 1.29% for the four scenarios shown in Fig. 6 (NRMSE values given by scenario from left to right). These values are lower than any other technique on average. These errors are due to the integration scheme with which we derive the FDT response function, as we only use a first-order integrator. Since it requires us to simulate two scenarios (one perturbed and one unperturbed), error can accumulate between these simulations; decreasing the integrator time
step or using a higher-order integrator (not shown) increases accuracy for this method. Despite this, the FDT gives us, up to the precision of our integrator, the system's true response function, which is a major advantage compared to the other techniques



which may or may not provide a physically-interpretable solution. The full implementation of the FDT requires a spatially explicit response matrix with multiple perturbation runs, but for a more even comparison to the other techniques, we only consider the well-mixed case here.

## 3.3    Experiment 2: Restricted Two Box Model

The restricted two box model investigates the impact of hidden variables (Fig. 1 (b)). This experiment is meant to test if an emulator can learn the true system response if not all information is included in the training data. Figure 7 summarizes the results of four emulation techniques (pattern scaling, deconvolution, modal fitting, and DMD) when trained and tested on different scenario combinations. Restricting the data means there is only one temperature series, rather than the three in the previous case. We therefore cannot calculate a global mean, and use a modified definition of pattern scaling in this section, mapping from forcing to temperature anomaly. As the FDT (Method II) has roughly equivalent performance to the previous section and is not impacted by the introduction of hidden variables, we omit it from this section.

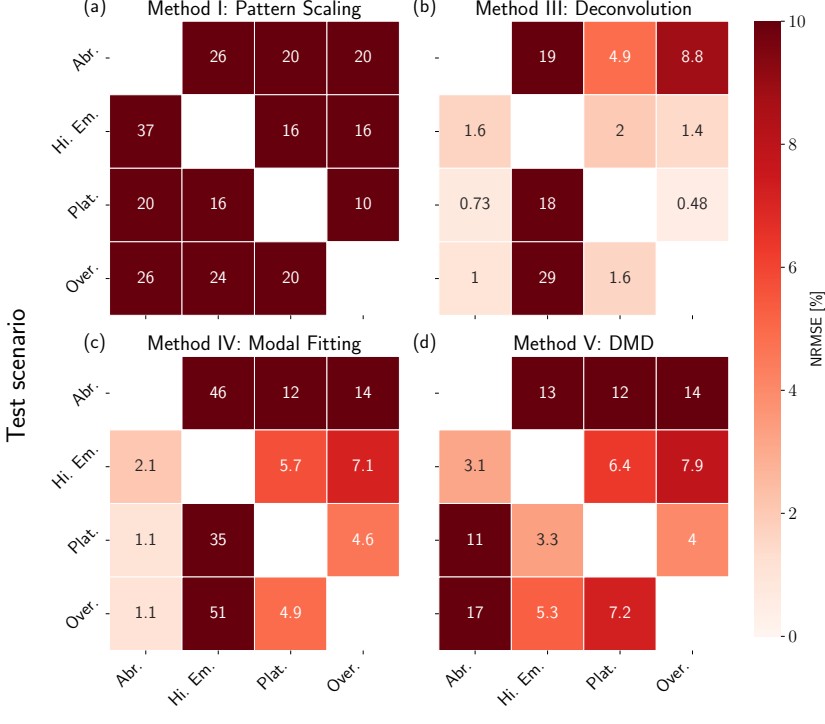

**Figure 7.** NRMSE heatmaps for pattern scaling (a), deconvolution (b), modal fitting (c), and DMD (d) emulators trained and tested against the restricted two box model. Results are shown in percentages, where lighter values correspond to lower error (higher performance) and darker values correspond to higher error (lower performance). Scenarios used for training are shown on the x-axis, while scenarios used for testing are shown on the y-axis. We do not include results for training and testing on the same dataset.



For all methods except deconvolution (Method III), we see a sharp drop in performance when introducing a hidden variable into the system. Deconvolution exhibits the same failure mode when training on *High Emissions* as before but to a greater

degree, along with the ill-conditioning failure mode when training on *Plateau* and *Overshoot*. Because this method treats each region as independent, it is more robust to the addition of hidden variables. It is able to capture the aggregate response of the atmospheric box that includes the influence of the ocean, but would not be able to separate those effects; i.e. the response function we derive is somewhat non-physical, though it can emulate the system effectively.

For the modal fitting emulator (Method IV), we initialize the optimization routine with guesses for both dominant modes

(the fast atmospheric response and slower oceanic response). It is largely unsuccessful in identifying these modes, except in the case of training with *Abrupt*. This scenario is unique in that both modes are visible in the atmospheric box alone (see the leftmost plot in the middle row of Fig. 3). Training on either *High Emissions* or *Overshoot* appears promising at first, but neither can extrapolate to *Abrupt*, meaning it effectively overfits on these scenarios and loses extrapolative capabilities. As before, we see that training on *High Emissions* leads to the worst performance overall, as this scenario is characterized by only

one effective timescale.

DMD (Method V) and by extension, EDMD (Method VI), experiences the sharpest decline in performance, with errors increasing by several orders of magnitude in some cases. Both methods see lower error in emulating scenarios similar to the training data (e.g. *High Emissions* vs. *Plateau*), but rapidly increasing error outside that regime. In addition to learning timescales like the previous two methods, DMD and EDMD are attempting to learning spatial interactions as well, meaning

they are disproportionately affected by the hidden variable. We can also frame this issue theoretically by stating that hidden variables violate one of the fundamental assumptions of E/DMD: the quantities we emulate are representative of all relevant system dynamics. By hiding the oceanic box, neither algorithm can learn the true physical behavior of the system. With EDMD, increases in polynomial order lead to further decreases in performance (not shown).

### 3.4 Experiment 3: Noisy Three Box Model

Results of the noisy three box model show how noise affects each emulator (Fig. 1 (c)). Figure 8 summarizes the results of four emulation techniques (pattern scaling, deconvolution, modal fitting, and DMD) when trained only on *Abrupt* and tested against the other three scenarios; we choose to train only on *Abrupt* as it yielded high performance across all methods (except pattern scaling) and we want to isolate the impact of noise. See Fig. 4 in Sect. 3.1 for performance metrics across all train/test combinations with a constant ensemble size. Since the noise is added linearly, taking the difference between the perturbed and

unperturbed ensembles effectively removes the noise when using the FDT (Method II). This leads to constant performance regardless of ensemble size, which is shown in Fig. 4. We additionally omit EDMD (Method VI) as it gives no improvements over DMD (Method V) in this linear case.

For these results, we evaluate performance relative to their noiseless baseline, rather than the absolute value of NRMSE; although *Abrupt* led to high performance for most methods, each method has a different baseline and some methods (e.g.

pattern scaling) performed poorly when trained on this scenario. All methods exhibit decreased performance in the noisy case relative to the noiseless baseline.





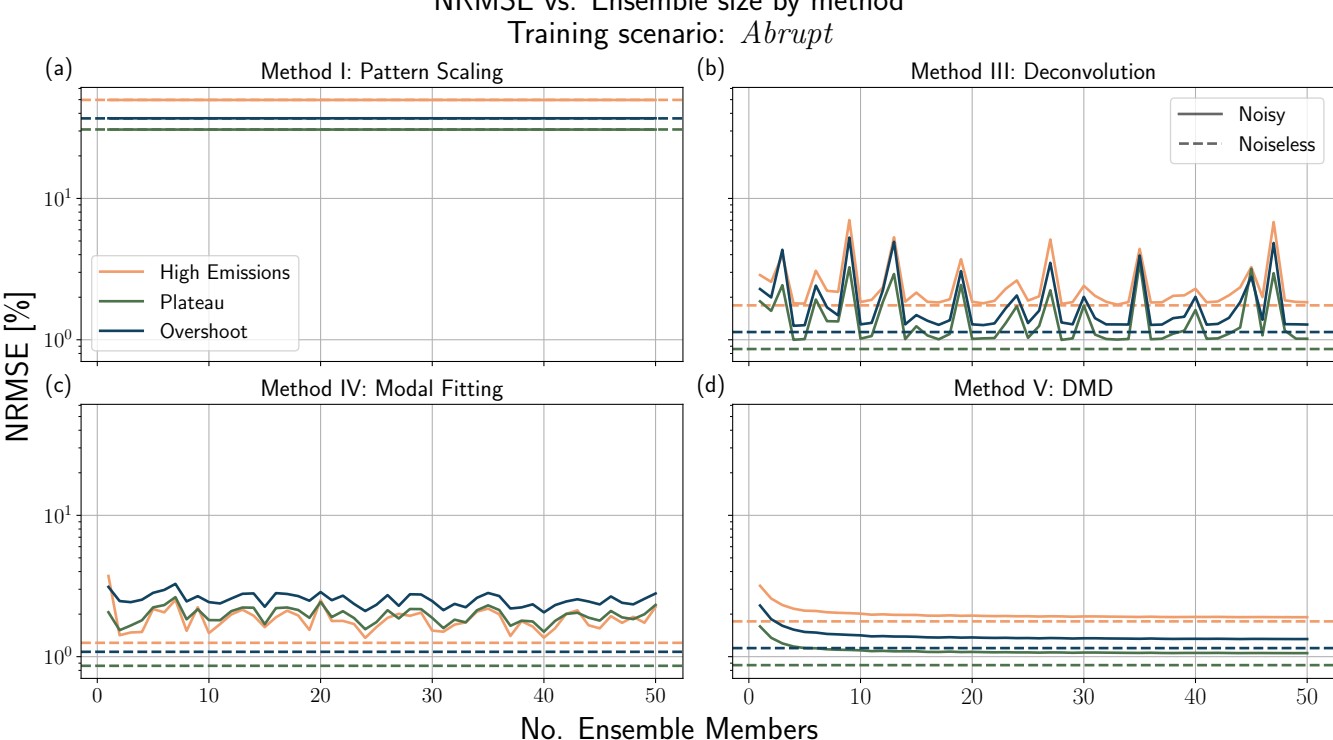

**Figure 8.** NRMSE vs. number of ensemble members for pattern scaling (a), deconvolution (b), modal fitting (c), and DMD (d) emulators trained on *Abrupt* and tested against the three remaining scenarios. Solid lines indicate the error in training/testing with noisy data, while the dashed lines indicate error in training/testing with noiseless data.

Pattern scaling (Method I) experiences no change in performance as the number of ensemble members is increased, as the linear regression smooths the data, reducing the impact of noise regardless of the ensemble size. With both deconvolution (Method III) and modal fitting (Method IV), there is an almost random change in performance depending on the number of
ensemble members. This is because both methods regularize the data. Deconvolution requires extra regularization when the system is noisy, or else the algorithm overfits on the noise, leading to extremely high error ($> \mathcal{O}(10^{10})$). The regularization has a similar effect to pattern scaling in making the expected performance of these algorithms more robust to noise. The variation in performance is due to the random sampling of ensemble members, with combinations that exhibit high error skewing the overall results. The error in DMD (Method V) is monotonically decreasing with ensemble size, though the presence of noise
leads to a drop in performance relative to the noiseless baseline.

### 3.5 Experiment 4: Cubic Lorenz System

The cubic Lorenz system allows us to jointly investigate the impact of chaos/noise and weak nonlinear effects on our emulators (Fig. 1 (c) and (d)). We run a 5,000 member ensemble as the variation in this experiment is much higher than the previous





noisy case. As in experiment two, we use a slightly modified definition of pattern scaling, mapping from forcing to quantity
of interest (the ensemble mean of $Z$). Figure 9 summarizes emulator performance against the number of ensemble members,
while Fig. 10 shows the response function derived using the FDT.

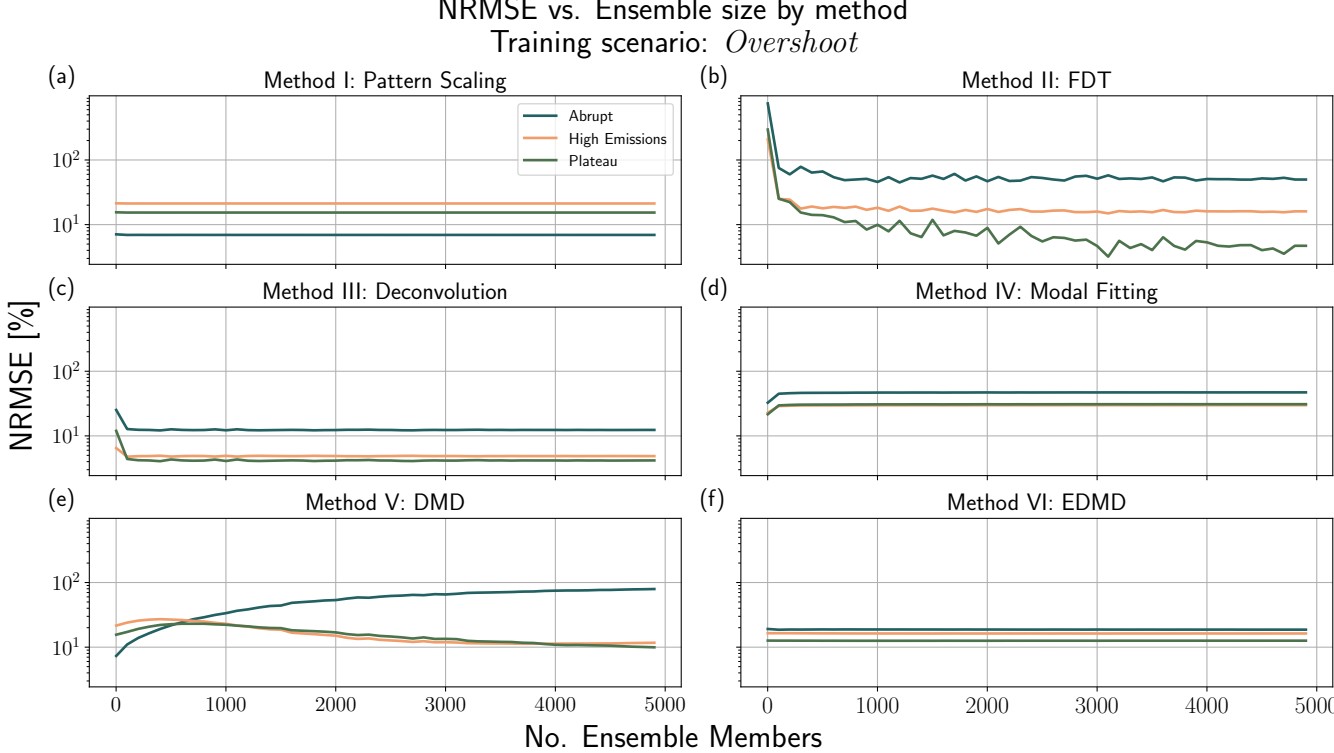

**Figure 9.** NRMSE vs. number of ensemble members for all emulators trained on *Overshoot* and tested against the three remaining scenarios. Emulators are shown as pattern scaling (a), FDT (b), deconvolution (c), modal fitting (d), and DMD (e), and EDMD (f). The FDT is trained on separate perturbation scenarios, and is therefore tested against all four scenarios. Unlike experiment three, there is no baseline/noiseless skill to compare against.

Similar to the previous noisy experiment (Sect. 3.4), pattern scaling (Method I) exhibits a constant level of performance independent of the number of ensemble members. The linear fitting process creates a strong artificial smoothing effect on the data, diminishing the potential impact of noise. This is also the case with both deconvolution (Method III) and the modal fitting
(Method IV) approach, both of which have little variability based on the number of ensemble members. The modal fitting approach additionally requires an imaginary component to enforce oscillations in the response function similar to those in the FDT result (Fig. 10). All approaches except DMD additionally show increased skill for smaller perturbations, i.e. higher skill in predicting *Plateau* than *Abrupt*. This is likely because smaller forcings lead to smaller deviations from the theoretical limit of response theory, which assumes small perturbations from the background state.





The performance of the FDT (Method II) is strongly dependent on the number of ensemble members. Figure 10 illustrates this point by showing how the response function derived using the FDT changes based on ensemble size. We treat the 50,000 member ensemble as our point of comparison, as further increases in ensemble size did not result in notable performance improvements. Key features, such as the initial magnitude of the response along with the time to reach that magnitude are consistent across all ensemble sizes, but the three cases deviate after this initial peak. All three cases exhibit a similar frequency

of oscillation over the time period tested, with noise in the 500 member ensemble influencing the longer-term behavior of that response (between years 3-5). There are deviations from the 50,000 member response in the 5,000 member case as well, though it is generally more in-phase than the 500 member ensemble. The NRMSE between the 50,000 and 5,000 member ensembles is 166.22%, while the NRMSE between the 50,000 and 500 member ensembles is 546.06%. Both responses are far from the ground truth, but the 5,000 member ensemble is much closer than the 500 member ensemble. Because the 5,000 member

ensemble has such high error relative to the 50,000 member ensemble, the predictive skill shown in Fig. 4 and Fig. 9 does not tell the full story. By further increasing ensemble size, we expect to see commensurate increases in accuracy when emulating this system with the FDT.

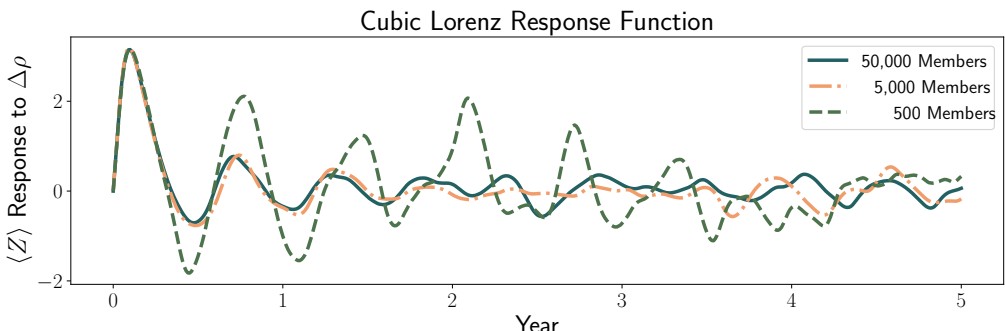

**Figure 10.** Response function for the cubic Lorenz system derived using the Fluctuation Dissipation Theorem with three sets of ensemble members: 500, 5,000, and 50,000. We use $\Delta t = 0.01$ and $\delta = 50\,\Delta t$ applied to the $Y$ component of the system.

Despite this experiment violating the linearity assumption of DMD (Method V), it has relatively stable performance of a similar order to the other methods tested. Predictive skill on *High Emissions* and *Plateau* increases with the number of ensemble

members, as one would expect as noise is averaged out, but skill on *Abrupt* decreases, which seems to be counterintuitive. In this case, we may not be introducing any further information about the coherent, underlying dynamics, which is supported by other methods showing consistent performance in these regimes. Increasing the ensemble size is leading to further refinement of the emulator's its parameters for *Overshoot* and its more closely related scenarios (*High Emissions* and *Plateau)*. A deeper investigation is required to assess DMD's suitability for Lorenz-like systems. EDMD (Method VI) does not exhibit this behav-

ior, instead performing with consistent skill across all combinations. This is likely because the 3rd-order Hermite polynomial used as the basis is well-suited to train on this scenario, illustrating the need for careful selection of basis functions.





## 4 Discussion and Conclusions

While emulators of Earth System Models (ESMs) have recently surged in popularity, uncertainty regarding their performance under a variety of scenarios and the lack of a comprehensive theoretical framework for analysis have posed problems for efforts at fundamental methodological comparisons. Our framework for emulator design and analysis builds on ideas from statistical mechanics and stochastic calculus, facilitating analysis of several emulation techniques from a theoretical and practical perspective. Our experiments based on simplified representations of the climate stress test a suite of emulators, including pattern scaling, response functions, and operator-based emulators, in the presence of memory effects, hidden variables, noise, and nonlinearities. Response function emulators consistently outperform other techniques, and the Fluctuation Dissipation Theorem (FDT) provides a robust method to derive them, though it also requires its own experimental ensemble. Table 6 summarizes the robustness of different emulators to different sources of error.

**Table 6.** Summary of emulator capability by technique based on the results from Sect. 3. An 'X' indicates a technique possess the listed capability, while a '∼' indicates may meet this requirement if other conditions are met; we discuss these capabilities explicitly in Sect. 4. *Memory* refers to an emulator's ability to capture memory effects (Fig. 1 (a), experiment one), *Hidden* refers to an emulator's skill in the presence of hidden variables (Fig. 1 (b), experiment two), *Noise* refers to an emulator's robustness to simulation noise (Fig. 1 (c), experiment three), and *Nonlin.* refers to an emulator's ability to capture weak nonlinear effects (Fig. 1 (d), experiment four).

| Technique | *Memory* | *Hidden* | *Noise* | *Nonlin.* |
|---|---|---|---|---|
| Method I: Pattern Scaling | | | X | |
| Method II: Fluctuation Dissipation Theorem | X | X | ∼ | ∼ |
| Method III: Deconvolution | X | X | ∼ | ∼ |
| Method IV: Modal Fitting | X | ∼ | X | ∼ |
| Method V: Dynamic Mode Decomposition (DMD) | X | | ∼ | |
| Method VI: Extended DMD | X | | ∼ | ∼ |

Each emulation technique considered in this work belongs to a spectrum of methods as defined by the joint Fokker-Planck/Koopman operator framework. Some emulators on this spectrum demand strict assumptions (quasi-linear/pattern scaling), while others are much more general (EDMD). There is a trade-off between the strictness of assumptions and emulator complexity, and relaxing these assumptions can shift the emulator's optimal use case. More general techniques may require specifically designed experiments, and decreasing structural emulator error may come at the price of increased computational costs (e.g. the Fluctuation Dissipation Theorem). Using this framework additionally identifies a gap in the current emulator





typology as defined by Tebaldi et al. (2025), as we need to consider the potential role operator-based emulators can play in this ecosystem; e.g. characterizing physical behavior in the system in addition to emulating it, as in Navarra et al. (2024).

Pattern scaling is a popular emulation technique because it is easy to implement, fast to apply, and its limits are well understood empirically (Mitchell, 2003; Tebaldi and Arblaster, 2014; Wells et al., 2023). Its efficiency makes it the method of choice particularly for assessments of mean annual temperature in monotonic forcing scenarios (e.g. SSP5-8.5, 3-7.0, or 2-4.5) and for understanding first-order trends of climate signals, even in the presence of internal variability. Previous work has shown this approach is valid only when the forcing is exponential and has a fixed spatial pattern, along with linear dynamics and

feedbacks (Giani et al., 2024). Our results additionally show that pattern scaling exhibits two sources of irreducible error: a mismatch between the true and predicted patterns at equilibrium and the assumption that the climate must respond instantaneously to external forcings. If forcing history is important, such as in centennial-scale or strong overshoot experiments, the single-pattern approximation breaks down, misrepresenting shifts in regional warming over time. This is also the case with highly variable fields such as precipitation, where the first-order approximation may not capture significant trends. More

general quasi-equilibrium approaches show promise (e.g. mapping from forcing to temperature in experiments two and four), but have yet to be widely explored in the context of full-scale ESMs. Pattern scaling's limitations push us towards emulation techniques that can capture more complex dynamics.

    Response functions are increasing in popularity as they can capture many processes of interest that are missed by pattern scaling, such as the pattern and memory effects (Freese et al., 2024; Sandstad et al., 2025; Winkler and Sierra, 2025; Womack

et al., 2025). This makes them ideal for representing decision-relevant, non-monotonic forcing scenarios, such as temperature overshoots. Response function approaches assume a linear relationship between the input forcing and output variable interest and that perturbations to the system are small (Lucarini et al., 2017). As a result, they are able to capture weakly nonlinear effects, so long as perturbations remain within the linear response regime. Response functions also have the potential to emulate longer (post-2100) scenarios due to their ability to capture memory effects, but longer ESM runs are required to test this. They

must be used with caution when nonlinear effects are dominant or (depending on the technique) when internal variability is significant.

    Despite its computational costs, deriving response functions with the Fluctuation Dissipation Theorem (FDT) offers a benefit over other response function techniques: it generates the system's exact linear response. Deconvolution and modal-fitting, by contrast, can produce non-physical output. As the FDT states, the response to small perturbations can be captured by $R(t)$ if

the system statistics are approximately stationary and the dynamics drive the weakly perturbed system back to the unperturbed state. The concept of climate is predicated on assuming the latter is true, further cementing the FDT's utility in this context. Because FDT-based response functions are physically interpretable, they support linear analyses of Earth system processes and serve as a reliable foundation for climate emulators (Lucarini and Chekroun, 2024). Although the approach requires large ensembles of non-standard perturbation experiments (Lucarini et al., 2017; Lembo et al., 2020), our results show that the

improvements in accuracy and interpretability may outweigh these costs, highlighting the need for further research on FDT-based emulation of full-scale ESMs.





Emulators that seek an explicit representation of the Koopman operator are potentially powerful tools as they are founded on rigorous theory and are interpretable (Tu et al., 2014; Williams et al., 2015; Schmid, 2022). They can, in principle, reproduce any behavior the climate system might exhibit. In practice, however, their utility is constrained by several factors. Both

Dynamic Mode Decomposition (DMD) and Extended DMD (EDMD) require the input and output variables of interest (e.g. radiative forcing and temperature) to completely characterize the dynamics of the system, rendering them sensitive to hidden variables. DMD additionally requires linearity between inputs and outputs, which is often violated in practice (Schmid, 2010). EDMD relaxes this assumption by using a higher-dimensional space at the cost of selecting an appropriate (and often problemspecific) set of basis functions (Williams et al., 2015). The choice of basis functions is a major consideration with this method,

and we may have been able to improve our implementation of EDMD further with a different choice. Solving the resulting large eigenvalue problems with either algorithm can be computationally demanding, and E/DMD can be sensitive to noise, potentially overfitting to data. Despite these challenges, operator methods allow us to identify dominant modes of variability in the climate system. They can also, in theory, be used to capture state-dependent and non-stationary processes, though this again requires a careful selection of basis functions and a large amount of training data. Further research is required to determine the

potential of operator-based methods for climate emulation.

Emulator performance varies depending on the experimental setup, highlighting that emulators are often designed to be application specific and not completely general. Figure 4 provides an overview of these results, but each emulator had the potential for high performance depending on the application. For example, pattern scaling performs poorly on all experiments, but shows high skill regardless of the experiment when trained and tested against *High Emissions*; this is not shown, as the

case where the training and testing datasets are the same is trivial (near zero error) for all emulation techniques. However, this illustrates that pattern scaling has utility if used on scenarios with exponential forcing, more akin to ScenarioMIP (O'Neill et al., 2016); see (Giani et al., 2024) for further discussion. Future work will further examine the role training data plays in emulator development.

Whether emulators learn physically interpretable representations of the system they are emulating remains an open question,

though our process of testing an emulator's extrapolative capability suggests that some techniques do learn the system's true behavior. The clearest example of this is the FDT, which performed consistently well across all scenarios. This is to be expected as the theory behind the FDT shows that it calculates the physical impulse response of the system (Lucarini et al., 2017; Giorgini et al., 2024). Pattern scaling on the other hand, by definition, does not learn realistic behavior unless the system is fully determined by the pattern scaling coefficients. For other techniques, the results are less clear. For example, the modal

fitting approach is able to extrapolate successfully in any of the first three experiments when trained on *Abrupt*, but not when trained on *High Emissions*, further supporting the need for an effort focused on quantifying the impact of training data on climate emulators. Deconvolution and DMD also exhibit mixed levels of extrapolative skill, leading to difficulties in making a consistent argument about interpretability from our results. This is especially the case for DMD, as the $\mathcal{L}$ matrix we derive is not easily mappable to the true underlying parameters of e.g. the coupled three box model, as this problem is effectively

underdetermined; we are solving for twelve DMD parameters, whereas the full system is determined by three heat capacities,



three feedback parameters, and one diffusion coefficient. Future work will investigate the possibility of learning true system parameters from these emulated representations.

Most work studying climate emulation focuses on the development and implementation of new approaches in an application-specific manner. Our results show the utility of an operator-based framework for systematic analysis and comparison across a spectrum of climate emulation techniques. The main benefit of this framework is that it provides a toolkit for understanding the trade-offs between emulator complexity and performance while connecting emulation techniques to fundamental principles of statistical mechanics and stochastic systems. Memory effects, internal variability, hidden variables, and nonlinearities are potential sources of emulator error, and response function-based emulators consistently outperform other methods, such as pattern scaling and Dynamic Mode Decomposition, across all experiments. Emulator performance varies by experimental setup, particularly through the choice of training data, and further work is required to fully characterize these effects in a rigorous way. This framework currently relies on simple experiments, and further work is needed to determine if operator-based methods like EDMD can be practically realized to emulate nonlinear processes in full-scale climate models. Our analysis also highlights the potential to use the Fluctuation Dissipation Theorem (FDT) to derive robust, physically-interpretable response functions to use for climate emulation, though this comes at the cost of increased computational resources. As interpretability is an ongoing point of discussion within the emulator community, investing resources in a physically-grounded method such as the FDT may go a long way towards increasing the utility of emulators not just for emulation, but for linear system analysis.

*Code and data availability.* All code to reproduce this work is available on github at https://github.com/cbwomack/SCM_Emulators (to be updated to Zenodo for publication). The raw data from CMIP6 were retrieved through the Earth System Grid Federation interface at https://aims2.llnl.gov/search/cmip6/. Figs. 2 - 10 were produced using scientific colour maps from Crameri (2023).

# Appendix A: Additional derivations

## A1  Pattern scaling errors

To understand the potential sources of error in pattern scaling, we start from the linear equation for temperature evolution

$$\frac{\partial}{\partial t}T(\boldsymbol{x},t) = \mathcal{K}(\boldsymbol{x},\boldsymbol{x}')T(\boldsymbol{x}',t) + P(\boldsymbol{x})F(t), \tag{A1}$$

where $T(\boldsymbol{x},t)$ is the spatially explicit temperature, $\mathcal{K}(\boldsymbol{x},\boldsymbol{x}')$ is the Koopman operator that governs the autonomous system dynamics, $P(\boldsymbol{x})$ is the spatial forcing pattern, and $F(t)$ is the time series of the forcing.

We can examine errors in pattern scaling by considering the case in which the pattern scaled temperature, $T_{PS}(\boldsymbol{x},t)$, is trained using an exponential forcing, $F(t) = e^{t/\tau}$, where $\tau$ indicates the growth rate of the exponential. Forcing our governing equation with this yields

$$T_{PS}(\boldsymbol{x},t) = \left[\frac{1}{\tau}\delta(\boldsymbol{x}-\boldsymbol{x}') - \mathcal{K}(\boldsymbol{x},\boldsymbol{x}')\right]^{-1} P(\boldsymbol{x}')e^{t/\tau}. \tag{A2}$$



Here $\delta(\boldsymbol{x} - \boldsymbol{x}')$ is the Dirac delta, so $\frac{1}{\tau}\delta - \mathcal{K}$ plays the role of $\frac{1}{\tau}I - \mathcal{K}$ in discretized form; we assume $\tau$ lies outside the spectrum of $\mathcal{K}$ so the inverse exists. Factoring out the exponential from this expression leaves us with

$$a_1(\boldsymbol{x}) = \left[\frac{1}{\tau}\delta(\boldsymbol{x} - \boldsymbol{x}') - \mathcal{K}(\boldsymbol{x},\boldsymbol{x}')\right]^{-1} P(\boldsymbol{x}'). \qquad (A3)$$

$a_1(\boldsymbol{x})$ is therefore the spatial scaling pattern used as our emulator. Inserting $T(\boldsymbol{x},t) = a_1(\boldsymbol{x})F(t)$ into the governing equation with the same exponential forcing, leaving us with

$$\frac{1}{\tau}a_1(\boldsymbol{x}) = \mathcal{K}(\boldsymbol{x},\boldsymbol{x}')a_1(\boldsymbol{x}') + P(\boldsymbol{x}). \qquad (A4)$$

This identity expresses how the pattern, $a_1(\boldsymbol{x})$, balances internal dynamics with an external forcing.

We now consider an alternate scenario with an arbitrary forcing, $F_{alt}$, that is not the exponential forcing used for training. We denote the error between the true solution and our emulator as

$$T'(\boldsymbol{x},t) = T_{alt}(\boldsymbol{x},t) - a_1(\boldsymbol{x})F_{alt}(t). \qquad (A5)$$

We then recognize that $T_{alt}(\boldsymbol{x},t) = T'(\boldsymbol{x},t) + a_1(\boldsymbol{x})F_{alt}(t)$. Inserting this into our governing equation and using the identity from Equation A4 gives an equation describing the evolution of errors over time

$$\frac{\partial}{\partial t}T'(\boldsymbol{x},t) = \mathcal{K}(\boldsymbol{x},\boldsymbol{x}')T'(\boldsymbol{x}',t) + \frac{1}{\tau}a_1(\boldsymbol{x})F_{alt}(t) - a_1(\boldsymbol{x})\frac{\partial}{\partial t}F_{alt}(t) \qquad (A6)$$

From this expression, we see that there are two distinct sources of error in pattern scaling when trained on an exponential (ScenarioMIP-like forcing). The first corresponds to an equilibrium-offset. If $F_{alt}(t)$ asymptotes to a constant $F_f$, the time

derivative in Equation A6 vanishes, leaving us with

$$\lim_{t \to \infty} T'(\boldsymbol{x},t) = -\frac{1}{\tau}\mathcal{K}^{-1}(\boldsymbol{x}',\boldsymbol{x})a_1(\boldsymbol{x})F_f. \qquad (A7)$$

Since we assume $\mathcal{K}^{-1}$ exists, there does not exist a non-zero vector such that $\mathcal{K}^{-1}(\boldsymbol{x}',\boldsymbol{x})a_1(\boldsymbol{x}) = 0$. Therefore the temperature produced by pattern scaling does not perfectly match the true equilibrium pattern.

The second source of error occurs in the transient case. When $F_{alt}(t)$ varies in time, the final term in Equation A6 does

not go to zero. If $F_{alt}(t)$ changes more quickly than the training growth rate (i.e. $\frac{\partial F_{alt}(t)}{\partial t} > \frac{1}{\tau}F_{alt}(t)$), then pattern scaling under-predicts the true temperature change. Conversely, very slow changes in $F_{alt}(t)$ lead to an over-prediction of the true temperature change. A non-negligible rate of change term signals that system memory will be significant in that scenario.

Physically, the first error arises because the system's equilibrium pattern depends on its slow internal modes, whereas the second arises because those modes cannot keep pace with forcing that accelerates faster (or slower) than the training rate $\tau$.

**A2   Deconvolution instabilities**

Deconvolution can amplify noise or in the worst case, cause the response function to blow up entirely. Here we identify where those instabilities arise. While issues with deconvolution are apparent in the time domain, they are easier to diagnose in



frequency space. We use the Fourier transform (denoted by $\mathcal{F}$) to rewrite convolution as multiplication:

$$\mathcal{F}[g(w_t)] = \mathcal{F}\left[\int_{-\infty}^{\infty} d\tau\, R(\boldsymbol{x},\tau)F(t-\tau)\right] \tag{A8}$$

$$\hat{g}(w_\omega) = \hat{R}(\boldsymbol{x},\omega)\hat{F}(\omega), \tag{A9}$$

where $g(w_t)$ is our statistical quantity of interest, $R(\boldsymbol{x},t)$ is the response function, $F(t)$ is the forcing, the hat denotes the (continuous-time) Fourier transform, and $\omega$ is the angular frequency. Recovering the response function therefore becomes division:

$$\hat{R}(\omega) = \frac{\hat{g}(w_\omega)}{\hat{F}(\omega)}, \tag{A10}$$

$$R(t) = \mathcal{F}^{-1}\left[\frac{\hat{g}(w_\omega)}{\hat{F}(\omega)}\right], \tag{A11}$$

where $\mathcal{F}^{-1}$ denotes the inverse Fourier transform. In discrete space, we use the fast Fourier transform.

If $\hat{F}(\omega)$ has any near-zero frequencies, dividing by it causes $\hat{R}(\omega) \to \infty$ at those frequencies. The corresponding time-domain process requires an explicit matrix inverse, where small eigenvalues translate into an ill-conditioned matrix. Additionally, if $|\hat{F}(\omega)|$ spans several orders of magnitude, the ratio $\hat{g}(w_\omega)/\hat{F}(\omega)$ amplifies high-frequency measurement noise and
round-off error. The condition number of the corresponding matrix becomes very large, yielding an unstable of estimate of $R(\boldsymbol{x},t)$.

These issues are also encountered in signal processing, where a system is said to lack a spectral inverse (i.e. zeros in the frequency domain) if it exhibits the above issues (Yeung and Kong, 1986; Zazula and Gyergyek, 1993). Even in the absence of noise, the relatively flat spectrum of a true impulse response makes it difficult to recover directly. A dominant eigenvalue can
obscure the weaker ones.

**A3 Distinction between Green's and response functions**

A scalar field, $w(t)$, governed by the linear time-invariant equation

$$\frac{\partial}{\partial t}w(t) = \mathcal{L}w(t) + F(t), \tag{A12}$$

has a corresponding Green's function, $G(t)$, that solves

$$\frac{\partial}{\partial t}G(t) = \mathcal{L}G(t) + \delta(t), \quad G(t<0) = 0. \tag{A13}$$

For a linear operator, $\mathcal{L}$, the solution is

$$G(t) = H(t)e^{\mathcal{L}t}, \tag{A14}$$





where $H(t)$ is the Heaviside step function and $e^{\mathcal{L}t}$ is a matrix exponential. From this, any general forcing produces a response given by

$$w(t) = \int_0^t G(\tau) F(t - \tau) \, d\tau. \tag{A15}$$

A response function, on the other hand, is either an empiral or equation-driven function that reproduces the system's linearized output but is not required to satisfy Equation A13. When the underlying dynamics are nonlinear, as is the case in climate models, a true Green's function does not exist. In practice however, the success of techniques such as pattern scaling illustrates that temperature response is very nearly linear for most of the globe, suggesting that data-derived response functions may closely approximate Green's functions for certain variables.

## A4 Transitioning from $\mathcal{K}$ to $\mathcal{L}$

We begin from a vector form of Equation 6, the expectation of a statistical field $g(\boldsymbol{w})$, where bold symbols are used to explicitly denote vectors. The vector $\boldsymbol{w}$ represents a set of state variables, $w_i$, at discrete points in space. The evolution of $\langle g(\boldsymbol{w}) \rangle$ is

$$\frac{\partial}{\partial t} \langle g(\boldsymbol{w}) \rangle = \left\langle \left[ \mathcal{N}_i(\boldsymbol{w}, t) + F_i(t) \right] \frac{\partial}{\partial w_i} g(\boldsymbol{w}) \right\rangle + D \left\langle \frac{\partial^2}{\partial w_i^2} g(\boldsymbol{w}) \right\rangle. \tag{A16}$$

We consider the case where $g(\boldsymbol{w}) = w_i$ to find the evolution of the mean of the state variables themselves. Substituting this gives

$$\frac{\partial}{\partial t} \langle w_i \rangle = \langle \mathcal{N}_i(\boldsymbol{w}, t) \rangle + F_i(t). \tag{A17}$$

We then define a steady baseline state, $\overline{\boldsymbol{w}}$, as

$$\langle \mathcal{N}_i(\overline{\boldsymbol{w}}, t) \rangle = -\overline{F}_i, \tag{A18}$$

where $\overline{F}_i$ is constant in time. Deviations from the baseline satisfy

$$\frac{\partial}{\partial t} \langle w_i' \rangle = \langle \mathcal{N}_i(\overline{\boldsymbol{w}} + \boldsymbol{w}', t) - \mathcal{N}_i(\overline{\boldsymbol{w}}, t) \rangle + F_i'(t), \tag{A19}$$

where $F_i'(t)$ is the time-varying component of the forcing. We then use a first-order Taylor expansion around $\overline{\boldsymbol{w}}$ to write

$$\frac{\partial}{\partial t} \langle w_i \rangle \simeq \left. \frac{\partial \mathcal{N}_i}{\partial w_j} \right|_{\overline{\boldsymbol{w}}} \langle w_j' \rangle + F_i'(t) = \mathcal{L}_{ij} \langle w_j' \rangle + F_i'(t), \tag{A20}$$

where the derivative term, $\frac{\partial \mathcal{N}_i}{\partial w_j}$, can be pulled out of the expectation because the baseline state is not stochastic. To conclude, we rewrite this with $\langle w_i' \rangle = T(x_i, t)$ and drop the discrete notation for space

$$\frac{\partial}{\partial t} T(\boldsymbol{x}, t) = \mathcal{L}(\boldsymbol{x}, \boldsymbol{x}') T(\boldsymbol{x}, t) + F(\boldsymbol{x}, t). \tag{A21}$$





## A5 FDT relationship to Fokker-Planck and Koopman

Here we show how the Fluctuation Dissipation Theorem (FDT) relates to the Fokker-Planck operator. The result shows that a linear response function can be computed directly from the forward operator of the unperturbed system.

Let $\mathbf{w}$ represent our full system state. Consider an equation of the form

$$\frac{\partial \mathbf{w}}{\partial t} = f_0(\mathbf{w}, t) + f_1(\mathbf{w}, t) + \varepsilon\xi(t), \tag{A22}$$

where $f_0$ governs the unperturbed system dynamics and $f_1$ governs the perturbed system dynamics. The Fokker-Planck equation corresponding to this is

$$\partial_t p + \nabla \cdot \left[ (f_0 + f_1)p - \frac{\varepsilon^2}{2}\nabla p \right] = 0. \tag{A23}$$

Without loss of generality, we decompose $p = p_0 + p_1$, where $p_0$ satisfies

$$\partial_t p_0 + \nabla \cdot \left( f_0 p_0 - \frac{\varepsilon^2}{2}\nabla p_0 \right) = 0. \tag{A24}$$

Then $p_1$ must exactly satisfy,

$$\partial_t p_1 + \nabla \cdot \left( f_0 p_1 + f_1 p_0 + f_1 p_1 - \frac{\varepsilon^2}{2}\nabla p_1 \right) = 0 \tag{A25}$$

The perturbation variables ($f_1$ and $p_1$) form a higher order term that we neglect, giving

$$\partial_t p_1 + \nabla \cdot \left( f_0 p_1 - \frac{\varepsilon^2}{2}\nabla p_1 \right) \approx -\nabla \cdot (f_1 p_0). \tag{A26}$$

The solution to this is

$$p_1(\mathbf{w}, t) = -e^{-\mathcal{F}_0 t}\nabla \cdot (f_1 p_0), \tag{A27}$$

assuming that $p_1(\mathbf{w}, 0) = 0$, i.e. there is no perturbation at $t = 0$, and $\mathcal{F}_0$ is the unperturbed (time-independent) Fokker-Planck operator. Multiplying through by an arbitrary statistical quantity of the state, $g(\mathbf{w})$, and integrating with respect to $\mathbf{w}$ then

yields the first order perturbation in $g(\mathbf{w})$

$$\int g(\mathbf{w})p_1(\mathbf{w}, t)\, d\mathbf{w} = \int g(\mathbf{w})e^{-\mathcal{F}_0 t}\nabla \cdot (f_1 p_0)\, d\mathbf{w}. \tag{A28}$$

The quantity on the left hand side is the expected value of the perturbed statistical quantity as a function of time. The right hand side is the cross correlation of the statistical quantity, $g$, with $h \equiv \nabla \cdot (f_1 p_0)/p_0$ with respect to the unperturbed system. Noting the Koopman operator is the adjoint of the Fokker-Planck operator gives

$$\left( e^{-\mathcal{F}_0 t} \right)^* = e^{-\mathcal{K}_0 t}, \tag{A29}$$

where $^*$ indicates the adjoint (conjuate transpose in finite dimensions) and $\mathcal{F}^* = \mathcal{K}$, giving an expression for the response function in terms of the Koopman operator.





Alternatively, we can connect the Fokker-Planck operator to the FDT through the score function. Consider the score function of the state given by

$$s(\mathbf{w}) = \nabla_{\mathbf{w}} \ln p_0(\mathbf{w}). \tag{A30}$$

For a small, instantaneous perturbation applied at $t = 0$, the linear response of the mean field at a lag $t$ is given by

$$R(t) = -\langle g(\mathbf{w}_t) s(\mathbf{w}_0) \rangle_{p_0}, \tag{A31}$$

where the angle brackets denote an average over the stationary ensemble. We express this correlation with a joint probability density as

$$R(t) = -\int\int p(\mathbf{w}_0, \mathbf{w}_t) g(\mathbf{w}_t) s(\mathbf{w}_0) \, d\mathbf{w}_t \, d\mathbf{w}_0, \tag{A32}$$

Using Bayes' theorem, we factor the joint probability density as

$$p(\mathbf{w}_0, \mathbf{w}_t) = p_0(\mathbf{w}_0) p(\mathbf{w}_t | \mathbf{w}_0), \tag{A33}$$

where $p(\mathbf{w}_t | \mathbf{w}_0)$ is the conditional probability from $\mathbf{w}_0$ to $\mathbf{w}_t$. For dynamics governed by the Fokker-Planck operator, $\mathcal{F}$, we have

$$p(\mathbf{w}_t | \mathbf{w}_0) = e^{\mathcal{F}t} \delta(\mathbf{w}_0 - \mathbf{w}_t). \tag{A34}$$

We then insert this expression into Equation A32 and integrate over $\mathbf{w}_t$:

$$R(t) = -\int p_0(\mathbf{w}_0) e^{\mathcal{F}t} g(\mathbf{w}_0) s(\mathbf{w}_0) \, d\mathbf{w}_0. \tag{A35}$$

Therefore, the linear response function can be obtained by propagating the unperturbed field with $e^{\mathcal{F}t}$ and correlating the result with the stationary score function.

**Appendix B: Regularization for response functions**

Estimating a response function from noisy data requires using deconvolution to invert an often ill-conditioned matrix. We choose to model the noise in our field of interest, $g(\mathbf{W})$, with a Gaussian noise term: $\varepsilon \sim \mathcal{N}(0, \sigma^2 \mathbf{I})$. Rather than applying an ad-hoc smoothing algorithm, we cast the problem in a Bayesian framework, placing a Gaussian prior on the response matrix: $\mathbf{R} \sim \mathcal{N}(0, \lambda^2 \mathbf{I})$. Our measurement model is therefore

$$g(\mathbf{W}) = \mathbf{FR} + \varepsilon. \tag{B1}$$

We have dropped $\Delta t$ and the spatial pattern for conciseness, but this analysis can easily be repeated including those terms.





Under this probabilistic model, we frame the task of estimating $\mathbf{R}$ as finding the vector that maximizes the response function probability given the observable data we have collected, i.e. $p(\mathbf{R}|g(\mathbf{W}))$. This term is called the *maximum a posteriori* (MAP). As it is more convenient to work with log probabilities, we recast this problem as

$$\max_{\mathbf{R}} \log p(\mathbf{R} \mid g(\mathbf{W})). \tag{B2}$$

Using Bayes theorem, maximizing the log-posterior,

$$\log p(\mathbf{R} \mid g(\mathbf{W})) = -\frac{1}{2\sigma^2} \|g(\mathbf{W}) - \mathbf{F}\mathbf{R}\|^2 - \frac{1}{2\lambda^2} \|\mathbf{R}\|^2 + \text{const}, \tag{B3}$$

is equivalent to solving

$$\min_{\mathbf{R}} \|g(\mathbf{W}) - \mathbf{F}\mathbf{R}\|^2 + \alpha \|\mathbf{R}\|^2, \quad \alpha = \sigma^2/\lambda^2. \tag{B4}$$

Thus ridge regression is equivalent to placing a Gaussian prior on the response function and assuming that the data we collect are corrupted by Gaussian noise.

To avoid making an arbitrary choice for our noise and prior variance hyperparameters parameters, $\sigma^2$ and $\lambda^2$, we propose to compute their maximum likelihood estimates under the distribution of the field of interest. We maximize the marginal likelihood evidence,

$$p(g(\mathbf{W}) \mid \sigma^2, \lambda^2) = \int p(g(\mathbf{W}) \mid \mathbf{R}, \sigma^2) p(\mathbf{R}, \lambda^2) \, d\mathbf{R} \tag{B5}$$

$$= \mathcal{N}\left(g(\mathbf{W} \mid 0, \boldsymbol{\Sigma}\right), \tag{B6}$$

with covariance $\boldsymbol{\Sigma} = \sigma^2 \mathbf{I} + \lambda^2 \mathbf{F}\mathbf{F}^T$. Maximizing the log-evidence,

$$-\frac{1}{2} \left( \log |\boldsymbol{\Sigma}| + g(\mathbf{W})^T \boldsymbol{\Sigma}^{-1} g(\mathbf{W}) \right) + \text{const}, \tag{B7}$$

has no closed-form solution for a general $\mathbf{F}$, so we determine $\sigma^2$ and $\lambda^2$ numerically.

## Appendix C: Analytic examples

In this appendix, we use a 1D Ornstein-Uhlenbeck (OU) process to analytically derive the Fokker-Planck operator, the Koopman operator, the eigenpairs of both operators, and the linear response function for the system obtained in two ways: (1) by directly solving the forced stochastic differential equation (SDE) and (2) by correlation with the score function.

### C1 Fokker-Planck and Koopman operator derivation

We define the OU SDE as

$$dw_t = -w_t dt + \sqrt{2} dW_t, \tag{C1}$$





where $w_t$ is the statistical field of interest and $W_t$ is a Wiener process. The drift coefficient, $-w_t$, relaxes the state toward zero, while the diffusion coefficient, $\sqrt{2}$, gives a unit variance.

We write the Fokker-Planck equation corresponding to this OU process directly:

$$\frac{\partial}{\partial t}p(w,t) = \frac{\partial}{\partial w}(wp) + \frac{\partial^2}{\partial w^2}p, \tag{C2}$$

where $p(w,t)$ is the probability density function of the field. The stationary solution of this expression is the standard normal probability density:

$$p_0(w) = \frac{1}{\sqrt{2\pi}}e^{-\frac{w^2}{2}}. \tag{C3}$$

From the previous result, we explicitly write the Fokker-Planck operator governing the evolution of the probability density as

$$\mathcal{F}(\cdot) = \frac{\partial}{\partial w}\left[w(\cdot) + \frac{\partial}{\partial w}(\cdot)\right]. \tag{C4}$$

To find the eigenfunctions, $\phi(w)$, with $\mathcal{F}\phi(w) = \lambda\phi(w)$, we introduce the ansatz $\phi(w) = h(w)e^{-\frac{w^2}{2}}$, giving

$$h''(w) - wh'(w) - \lambda h(w) = 0, \tag{C5}$$

whose solutions are Hermite polynomials, $H_n(w)$, with eigenvalues $\lambda = -n$ for $n = 0, 1, 2, \ldots$.

## C2  Response function via direct diagnosis

Adding a deterministic forcing, $F(t)$, to our OU process gives

$$dy_t = (-y_t + F(t))dt + \sqrt{2}dW_t. \tag{C6}$$

Taking the expected value of this and assuming $\langle y(0) \rangle = 0$ gives

$$\frac{d}{dt}\langle y \rangle = -\langle y \rangle + F(t), \tag{C7}$$

whose solution is given by

$$\langle y(t) \rangle = \int_0^t e^{-\tau}F(t-\tau)\,d\tau, \tag{C8}$$

where the response function is $R(t) = e^{-t}$ for $t \geq 0$.

## C3  Response function via correlation with score function

The stationary score function is given by

$$s(w) = \nabla_w \ln p_0(w) = -w, \tag{C9}$$



where we can make this simplification since the stationary probability distribution is given by a standard normal.

The Fluctuation Dissipation Theorem predicts

$$R(t) = -\langle w(t)s(w(0))\rangle = e^{-t}, \tag{C10}$$

which agrees exactly with the direct solution above.

*Author contributions.* Conceptualization: CW, GF, SB, SE, NS. Formal analysis: CW, GF, SB, AS. Funding acquisition: NS. Investigation: CW. Methodology: CW, GF, SB, AS, PG. Supervision: GF, SE, NS. Visualization: CW, SB. Writing - original draft: CW, GF, SB. Writing - reviewing and editing: CW, GF, SB, AS, PG, SE, NE.

*Competing interests.* The authors declare that they have no conflict of interest.

*Acknowledgements.* This research was part of the Bringing Computation to the Climate Challenge (BC3) project and supported by Schmidt
Sciences, LLC. through the MIT Grand Challenges. We also acknowledge the MIT *Svante* cluster supported by the Center for Sustainability Science and Strategy for computing resources. We are grateful for the entire BC3 team who provided insightful feedback and discussions about this work. We acknowledge the World Climate Research Programme, which, through its Working Group on Coupled Modelling, coordinated and promoted CMIP6. We thank the climate modeling groups for producing and making available their model output, the Earth System Grid Federation (ESGF) for archiving the data and providing access, and the multiple funding agencies who support CMIP6 and
ESGF. We additionally acknowledge the use of LLMs in editing and annotating the code associated with this work.



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
