# Peer review of "A theoretical framework to understand sources of error in Earth System Model emulation"

_EGUsphere, 2025_

## Referee Comment (RC2)

The authors make both conceptual and methodological contributions to the emulator literature. Conceptually, they frame the climate system as a stochastic process whose evolution can be described either through probability densities or statistical moments, using operator theory and linear response. A key contribution is their demonstration of how different emulation techniques connect to operator and response frameworks, highlighting a current gap: the underexplored role of operator-based emulators. Methodologically, they implement six emulation techniques and evaluate their skill on toy climate models designed to test memory effects, hidden variables, internal variability, and nonlinearities.

Overall, I find the contribution meaningful. Emulator development has so far been largely driven by immediate data needs, and the field lacks a unifying conceptual baseline. This paper takes an important step toward such a baseline and clarifies how approaches from other disciplines fit into the emulator literature. I also find the discussion of pattern scaling valuable, as it helps delineate the conditions under which this widely used technique succeeds or fails.

I see some room for improvement, particularly regarding structure and clarity. Below, I organize my comments along three dimensions: the theoretical framework, the connection between emulators and theory, and the experimental design. Each section starts with a summary of what I think the paper said followed by specific comments. Feel free to use (or ignore) everything as you see fit.

**Theoretical groundwork**

The paper frames the climate system as a stochastic process whose evolution over time can be described using stochastic differential equations (SDEs). This description allows the dynamics to be restated in terms of operators, which can act either on the probability density (Perron–Frobenius/Fokker–Planck operator) or on observables (Koopman operator). Observables include, but are not limited to, the moments of the distribution (mean, variance, etc.). The two operators are dual: the Fokker–Planck view evolves the density and integrates against the observable to obtain expectation values, while the Koopman view evolves observables directly, with expectations computed with respect to the initial distribution. The operator view allows the problem to be restated as an eigenproblem (e.g., Fokker–Planck view = eigenfunctions are modes of probability densities and eigenvalues represent decay rates). And linear response theory is essentially about considering perturbations to they system and expressing them in terms of solutions to the unperturbed system. More general than the fluctuation-dissipation theorem (FDT) presented in the paper, would be Ruelle's response theory (not mentioned) as it applies to chaotic, non-equilibrium systems. I am not an expert on SDEs, but found the concepts familiar from quantum mechanics (Schrödinger picture vs. Heisenberg picture, Hamiltonian generators, perturbation theory).

**Comments / Suggestions:**
1. **Framing & readability**: I found the theoretical framing very helpful and intuitive, especially the explanations around L.145 ff. To make it more accessible, I suggest separating the conceptual groundwork from the emulation idea:
   - Introduce a dedicated "Theoretical Foundation" section to present the stochastic framework and operator duality independently of the emulation

target. I think this would help establishing a common baseline first and to invite readers to only read the part they are interested in.
   - Then have a section "Connecting Emulators to Theory" where you introduce what an emulator is meant to do (current 2.2).
   - Finally, present "Experiments/Applications" (current 2.3 + 3).
2. **SDE context**: As I understood, Eq. 1 is a special case derived by Hasselmann under assumptions such as time-scale separation between fast (weather) and slow (climate) variables, stationarity of the fast system, and viewing fast processes as random forcing. A brief explanation of these conditions and the meaning of each term (e.g., the white noise term representing aggregated fast-variable effects) would aid clarity.
3. **Equation consistency**: In Fig. 2 you write $\frac{\partial w}{\partial t} = \mathcal{N}(w, F) + \epsilon\xi$ while Eq. 2 is $\frac{\partial w}{\partial t} = \mathcal{N}(w) + F(t) + \epsilon\xi$. These are not equivalent if feedback parameters depend on forcing. I don't quite understand which formulation is used when and why?
4. **FDT introduction**: Before directly introducing linear response theory via the FDT (L.219 ff.), an intermediate step would help:
   - Define an unperturbed generator $K_0$ and perturbation, $\delta K$, so $K = K_0 + \delta K$
   - Then the perturbed expectation value of an observable g is $\langle g \rangle_t^{perturbed} = \langle g \rangle_t^{(0)} + \delta\langle g \rangle_t + \mathcal{O}(\delta^2)$.
   - From there, Ruelle's response theory provides the general solution, with FDT as the special equilibrium case (Eq. 12)

**Connecting emulators to the theoretical groundwork**

Earth system models generate data that implicitly obeys Eq. 1 (or its operator-based equivalents), but they do not provide us with the exact operators or their solutions. Instead, we observe samples, and the emulator's task is to approximate either the full distribution or selected observables from these samples. The key conceptual contribution of the paper is to connect emulator-based approaches to operator and response formulations. Modern probabilistic models such as Bayesian inference or diffusion models are naturally connected to the state-based view (Fokker–Planck). For example, diffusion models learn a score function (the gradient of the log density), which directly appears in the Fokker–Planck operator. From this perspective, such emulators can be seen as approximating the Fokker–Planck operator, and thus as providing an entry point to linear response theory: score-based response functions are essentially obtained by applying linear response but replacing the Fokker–Planck operator with the learned score. Similarly, emulators that target specific observables (e.g., the mean of the distribution) connect to the Koopman operator and can therefore be framed within linear response theory as well. The authors explicitly work out these connections for six emulation techniques, focusing on temperature anomalies, and discuss the errors each emulator makes in approximating the underlying operators.

**Comments / Suggestions:**
1. **Clarity and structure:**
   - This conceptual bridge between emulators and operator theory is central to your paper. I recommend making it an independent section that explicitly highlights

these links and their implications for emulator errors (see comment on Framing in the previous section)
- An additional table or expanded version of Table 1 would be very helpful. It could summarize pros/cons of each approach, e.g. the computational speed of pattern scaling vs. its structural biases, or the expressiveness of score-based emulators vs. challenges in accessibility and training. In addition, the conncetion of Table 1 to Fig. 2 could be strengthened by adding a column called Emulator type (or adding brackets) that show Method I belongs to Pattern Scaling + Extensions; Method II + IV are impulse response emulators and Method V + VI to Operator-Based emulation
- Figure 1 and Appendix A1 are excellent in motivating the error sources and in giving an example of how your framework helps identifying them .

2. **Data expectations**: When first reading Section 2.2, I expected the framework to be applied to CMIP6 data (reinforced by Fig. 1). It only became clear later.

3. **Assumptions in pattern scaling vs. impulse response vs. operator-based modelling:**
- In L. 268 & 299 ff: you righty point out that pattern scaling assumes time-invariance and quasi equilibrium and then mention equilibrium conditions in L. 314 ff. again in terms of FDT. I found the mentioning of two types of equlibrium conditions a bit fuzzy upon first reading and I think it wuld make sense to be a bit more explicit
- In L. 440 ff. you introduce operator-bsed emualtors as the most general class. This makes sense because the previously introduced emulators have some equilibrium assumptions. I feel like the generalisability of this emulator-based framework could be highlighted a bit more; consider making the assumptions in Fig. 2 (arrow from 3b to 3d) more explicit

4. **Conclusions:** Your reflections in L. 838 ff. are supper fitting. For me, the theoretical contributions were the most compelling part of the paper, since many existing studies implement emulators more naively. As you argue, the lack of a conceptual baseline makes it hard to integrate insights across disciplines, and I would encourage you to highlight this contribution more strongly throughout the manuscript.

**Experimental approach (sources of error)**

The authors employ four simple climate models (a two-box model, a three-box model, a noisy box model, and a cubic Lorenz system) and drive each with four structurally distinct forcing pathways (abrupt, transient high-emissions, plateau, and overshoot). For each experiment, they train their emulators on data from one scenario and test them against all other scenarios, thereby comparing performance across settings. The experimental design is deliberately simple: it targets errors arising from dynamical features such as memory, hidden variables, noise, and nonlinearities, rather than errors linked to spatial heterogeneity. This choice has clear advantages—the box-model experiments are transparent, reproducible, and well-suited for stress-testing emulator failure modes—but it also carries disadvantages, as spatial patterns are not represented and emulator performance is reduced to a single aggregated metric across boxes.

**Comments / Suggestions:**

1. **Data expectations**: Reiterating the previous point. Initially, I found the experimental set-up somewhat confusing, as I expected the tests to involve spatially resolved ESM data. The title, abstract, and introduction explicitly mention ESMs; Fig. 1 also presents spatially resolved data; and in Section 2.2 the term "spatial" led me to expect an evaluation of spatial error. In practice, however, the experiments are based on box models with at most three degrees of "spatial" resolution. Applying your framework to toy models is, in my view, valuable—as it provides a controlled setting for isolating and examining specific effects—but I think this would be clearer if explicitly framed as a proof-of-concept.

2. **Averaging of results**: The data only ever shows a single evaluation score, while sometimes the models have multiple boxes. Do you average across boxes?

3. Table 6 summarizes the experimental findings well.

4. I appreciated the conclusions in L. 816 ff.

**Other minor suggestions**

- Fig. 2:
    - Add a reference to Table 1 to the description (helpful for understanding the boxes on the right given you refer to Fig.2 already in L.77, but mention the Methods I-VI only from L.115 onwards)
    - What is the difference between solid and dashed arrows (e.g., going from 2b to 3b as opposed to 3b to 3d)?
- L. 759: Non-linearities as opposed to nonlinearities throughout the remainder of the manuscript

---

## Author Comment (AC1)

Author response to RC1 for "A framework for assessing and understanding sources of error in Earth System Model emulation"

By Christopher B. Womack et al.

We first thank the two anonymous reviewers for their constructive comments and careful engagement with our substantial manuscript. Their effort has helped us improve the framing, quality, and clarity of our contribution.

We now respond to RC1, which is reproduced in black text below. Our responses follow immediately in red text, and any additions to the manuscript are included in italic red text.

**Anonymous Referee #1**

The authors present a framework for comparing emulation techniques. They do so by showing the theoretical connections between several existing emulation methods and relating them to two types of linear operators. These operators are shown to explain the same information about the system, demonstrating a link among all methods considered. The authors then test these methods' abilities to predict four forcing response scenarios in four simplified toy models of either the climate system or the Lorenz convection approximation. Response function methods outperform both pattern scaling and attempts to directly estimate the linear operator in these example tests. The discussion around modeled results in the various tests is thorough and the connections to a common set of linear operators will likely be useful when considering how different emulators might perform. I have experience with pattern scaling, FDT, and ridge regression (which is how the deconvolution method has been practically implemented), though less so with much of the emulator-specific background cited here. As such, I will limit my comments to how this work fits with understanding ESMs more broadly.

**Specific comments:**

My main comment covers the goal and applicability of this work. I understand that the intent of the paper is to establish a "framework", by which the authors mean the ability to frame each of these emulators as a variation or simplification on the paired linear response operators Fokker-Planck/Koopman. What is less clear to me is how directly the link can be made to "sources of error in Earth System Model emulation". Generally, I understand if this paper is laying the groundwork for ESM testing, but in that case I felt that the writing did not make that intention clear. As presented, it reads as offering a tool that is directly applicable to evaluating emulators with respect to ESMs. The tests get at particular challenges in ESMs: memory effects, hidden variables, noise, and

nonlinearities. However, the reader does not see the actual interaction between these methods and errors in ESM emulation.

Based on this and comments from Reviewer #2, we agree the manuscript has a framing issue relative to its treatment (or lack thereof) of ESMs. The value in the theoretical framing is that we can use it to assess and improve emulators by analyzing where errors arise. We can evaluate assumptions present in many common emulation techniques, along with what types of error those assumptions invite and how this is problematic for ESMs. To help clarify these points, we will restructure the manuscript slightly, separating the theory (now Sect. 2) from our simplified experiments (now Sect. 3 + results in Sect. 4). We will also make the following changes to our abstract and introduction to clarify our experimental setup, along with other minor changes throughout to ensure continuity with these structural changes.

Addition to abstract: To support our theoretical contributions, we provide practical implementation details for each technique, along with discussion on the relative utility of these emulation methods. We evaluate emulator performance using simplified climate models, including box models and a modified version of the Lorenz 63 model, across a series of experiments designed to highlight different potential sources of error.

Changes to introduction (final paragraph): Section 2 first presents our theoretical framework, highlighting that the goal of many emulation techniques is to simplify complex climate dynamics into a linear set of modes associated with the Fokker-Planck and Koopman operators. We then apply this framework to identify potential sources of error within six emulation techniques, analyzing them from both a theoretical and practical perspective (Sect. 2.3). In Sect. 3, we introduce a series of experiments using simplified climate models and forcing scenarios designed to stress test and evaluate each emulator; these experiments include box models and a modified version of the Lorenz 63 system. Section 4 contains experimental results, showing that response functions consistently outperform other emulators across potential high-error scenarios. We conclude by discussing optimal use cases for each emulator, along with implications for ESMs based on our pedagogical model results (Sect. 5).

The reviewer is correct in that we do not emulate ESMs in this work. We will add an "Implications for ESMs" subsection in the discussion to explicitly address the utility of our framework in that context.

Implications for ESMs: While the lack of a common conceptual baseline has historically hindered comparisons between emulator classes, our framework takes an important step towards resolving this. Efforts such as ClimateBench, which provide a common

training and evaluation benchmark, have been useful to that end (Watson-Parris et al., 2022), but emulator structural differences prevent this framework from being applied to all existing emulation techniques. Additionally, the high computational burden of running scenarios beyond those in the CMIP archive (for training or evaluation), prevents rigorous assessment of emulator capability (e.g., emulating the impact of individual forcings) and generalizability (accuracy beyond ScenarioMIP). Results from experiments such as the Detection and Attribution MIP (DAMIP) and Regional Aerosol MIP (RAMIP) can help fill these gaps (Gillett et al., 2016; Wilcox et al., 2023), but the field of ESM emulation is currently data-constrained. Our theoretical framework provides value in this data-limited setting, as it allows us to evaluate the assumptions present in many common emulators. Our results illustrate the potential sources of error different emulator structural assumptions invite, giving us tools to assess and improve emulation techniques independently of ESM results. As ESM outputs improve with CMIP7 and beyond, this framework can help ensure emulators are prepared to train on those new results.

Our pedagogical experiments provide a useful tool to isolate and examine individual sources of error when emulating ESMs (Fig. 1). Though our simplified models are limited in that they lack much of the complexity of full-scale ESMs, our experiments highlight that emulator errors can be proactively resolved through structural changes, regardless of the parent model. For example, our results further support the growing body of literature on the utility of response functions (Freese et al., 2024; Womack et al., 2025; Winkler and Sierra, 2025). Response functions offer improvements over traditional pattern scaling, particularly when considering memory effects in decision-relevant scenarios. They may also emulate longer (post-2100) scenarios by accounting for regional pattern shifts, though longer ESM runs, such as the extensions proposed in ScenarioMIP for CMIP7, are required to test this (Van Vuuren et al., 2025). Existing emulators of ESMs may also benefit from incorporating response functions, c.f., recent work into hybrid emulation using generative machine learning methods in addition to pattern scaling (Bouabid et al., 2025).

Several promising emulation techniques explored here, including the Fluctuation Dissipation Theorem (FDT), Dynamic Mode Decomposition (DMD), and Extended DMD (EDMD), have seen uses in climate science but have yet to be applied directly as emulators of ESM outputs as defined by Tebaldi et al. (2025). An intermediate step for either the FDT or EDMD and DMD may be to first emulate an EMIC, helping determine useful training scenarios without the cost of a full ESM. Our results suggest further research into these techniques is warranted, as they may represent more complex dynamics than other methods. In this context, the FDT stands apart as the most promising technique for emulating general dynamical systems, as evidenced by its skill

in this and other recent work (Giorgini et al., 2025b). However, using the FDT to derive response functions through perturbations requires a full initial condition ensemble for every perturbed grid cell/region (Lucarini et al., 2017; Lembo et al., 2020), similar to the Green's Function MIP (Bloch-Johnson et al., 2024), and is likely prohibitively expensive for full ESMs. The score-based FDT (Sect. 2.3) provides a remedy, using statistical learning methods to learn the score function and thus the system response (Giorgini et al., 2025b). Regardless of the derivation method, our results suggest response functions are the dominant emulation technique both in terms of accuracy and interpretability.

Most work studying climate emulation focuses on developing and implementing new approaches in an application-specific manner. Our results show the utility of an operator-based framework for systematic analysis and comparison of climate emulation techniques. The main benefit of this framework is providing a toolkit for understanding trade-offs between emulator complexity and performance while connecting emulation techniques to fundamental principles of statistical mechanics and stochastic systems. We find that memory effects, internal variability, hidden variables, and nonlinearities are potential error sources, and that response function-based emulators consistently outperform other methods, such as pattern scaling and DMD, across all experiments. Emulator performance varies by experimental setup, particularly through the choice of training data, and further work is required to fully characterize these effects. This framework currently relies on simple experiments, and further work is needed to determine if operator-based methods like EDMD can be practically realized to emulate nonlinear processes in full-scale climate models. Our analysis also highlights the FDT's potential for deriving robust, physically-interpretable response functions, though its computational cost is a potential barrier. As interpretability is an ongoing discussion in the emulator community, investing resources in physically-grounded methods like the FDT may go a long way towards increasing the utility of emulators not just for emulation, but for linear system analysis.

530: While the 2- and 3-box models are frequent approximations to the climate system, they lack many of the physical mechanisms that make the climate system difficult to model. The parameters in these models are fit to ESMs, so are themselves simplified estimates of the actual behavior. I felt that the link between ability to emulate these examples and the ability to emulate ESMs deserved more discussion. I would have found this conceptually more useful than the level of technical detail included for the linear operators and each emulation model in the main text.

On the limitations of box models, we agree that these lack many of the physical mechanisms present in full-scale climate models and that is a limitation of this work. We will add discussion around this point to the Implications for ESMs section.

From paragraph two of Implications for ESMs: Our pedagogical experiments provide a useful tool to isolate and examine individual sources of error when emulating ESMs (Fig. 1). Though our simplified models are limited in that they lack much of the complexity of full-scale ESMs, our experiments highlight that emulator errors can be proactively resolved through structural changes, regardless of the parent model.

On the applicability of this methodology to ESMs, we agree that a more explicit discussion around this topic is necessary for this manuscript. The additional Implications for ESMs subsection covers these points more explicitly.

846: "This framework currently relies on simple experiments, and further work is needed to determine if operator-based methods like EDMD can be practically realized to emulate nonlinear processes in full-scale climate models.": this sentence to me suggests that the step of showing that this framework is useful for ESMs is left to future work. I can see that there is some value in being able to connect the different models through a common framework in the way the authors use it to diagnose differences in the toy model. This may be more in line with a proof of concept for the framework rather than demonstrating how the framework applies to ESMs. However, if the goal is for this framework to be used by others and applied to ESMs, this seems like an important step to include. This may just be a framing issue.

We agree that this first draft suffers from a framing issue. While we do not apply our framework directly to ESMs in this manuscript, formalizing these ideas through our idealized experiments constitutes necessary foundational work towards that goal. The previous structural changes to the manuscript will help highlight the utility of our contribution.

From paragraph one of Implications for ESMs: Our theoretical framework provides value in this data-limited setting, as it allows us to evaluate the assumptions present in many common emulators. Our results illustrate the potential sources of error different emulator structural assumptions invite, giving us tools to assess and improve emulation techniques independently of ESM results. As ESM outputs improve with CMIP7 and beyond, this framework can help ensure emulators are prepared to train on those new results.

Figure 4: If the results suggest that directly estimating response operators is the most prone to error, does this challenge the response operator framework as the most useful common link for the different emulation methods? This seems to suggest the Koopman operator is not the most useful simplification of the climate system.

This is a great observation, as directly estimating these operators is a nuanced, data-intensive task. While EDMD and DMD attempt to learn the Koopman operator, they are extremely simplified representations and in most cases do not closely approximate the true operator. Despite this, the Koopman and Fokker-Planck operators provide the most useful theoretical basis as they offer a way to directly link vastly different forms of emulators. We will add text to clarify the differences between the theoretical and data-derived Koopman operators in the discussion.

Addition to discussion: While EDMD and DMD attempt to approximate the Koopman operator, they are simplified representations and in many cases do not closely approximate the true operator. Despite this, the Koopman and Fokker-Planck operators provide the most useful theoretical basis as they offer a way to directly link disparate forms of emulators.

**Minor technical:**

42: "Impulse response (response/Green's function) methods" this wording is confusing, how is "response" an example of "impulse response"

We agree, the original wording here was unclear. We will clarify the intended meaning of this phrase.

Change to introduction: Impulse response methods, commonly referred to as either response or Green's functions,...

---

## Author Comment (AC2)

Author response to RC2 for "A framework for assessing and understanding sources of error in Earth System Model emulation"

By Christopher B. Womack et al.

We first thank the two anonymous reviewers for their constructive comments and careful engagement with our substantial manuscript. Their effort has helped us improve the framing, quality, and clarity of our contribution.

We now respond to RC2, which is reproduced in black text below. Our responses follow immediately in red text, and any additions to the manuscript are included in italic red text.

**Anonymous Referee #2**

The authors make both conceptual and methodological contributions to the emulator literature. Conceptually, they frame the climate system as a stochastic process whose evolution can be described either through probability densities or statistical moments, using operator theory and linear response. A key contribution is their demonstration of how different emulation techniques connect to operator and response frameworks, highlighting a current gap: the underexplored role of operator-based emulators. Methodologically, they implement six emulation techniques and evaluate their skill on toy climate models designed to test memory effects, hidden variables, internal variability, and nonlinearities.

Overall, I find the contribution meaningful. Emulator development has so far been largely driven by immediate data needs, and the field lacks a unifying conceptual baseline. This paper takes an important step toward such a baseline and clarifies how approaches from other disciplines fit into the emulator literature. I also find the discussion of pattern scaling valuable, as it helps delineate the conditions under which this widely used technique succeeds or fails.

I see some room for improvement, particularly regarding structure and clarity. Below, I organize my comments along three dimensions: the theoretical framework, the connection between emulators and theory, and the experimental design. Each section starts with a summary of what I think the paper said followed by specific comments. Feel free to use (or ignore) everything as you see fit.

**Theoretical Groundwork**

The paper frames the climate system as a stochastic process whose evolution over time can be described using stochastic differential equations (SDEs). This description allows the dynamics to be restated in terms of operators, which can act either on the probability density (Perron-Frobenius/Fokker-Planck operator) or on observables (Koopman operator). Observables include, but are not limited to, the moments of the distribution (mean, variance, etc.). The two operators are dual: the Fokker–Planck view evolves the density and integrates against the observable to obtain expectation values, while the Koopman view evolves observables directly, with expectations computed with respect to the initial distribution. The operator view allows the problem to be restated as an eigenproblem (e.g., Fokker-Planck view = eigenfunctions are modes of probability densities and eigenvalues represent decay rates). And linear response theory is essentially about considering perturbations to they system and expressing them in general than the of solutions to the unperturbed system. More fluctuation-dissipation theorem (FDT) presented in the paper, would be Ruelle's response theory (not mentioned) as it applies to chaotic, non-equilibrium systems. I am not an expert on SDEs, but found the concepts familiar from quantum mechanics (Schrödinger picture vs. Heisenberg picture, Hamiltonian generators, perturbation theory).

**Comments / Suggestions:**

- 1. **Framing & readability:** I found the theoretical framing very helpful and intuitive, especially the explanations around L.145 ff. To make it more accessible, I suggest separating the conceptual groundwork from the emulation idea:
- Introduce a dedicated "Theoretical Foundation" section to present the stochastic framework and operator duality independently of the emulation target. I think this would help establishing a common baseline first and to invite readers to only read the part they are interested in.
- Then have a section "Connecting Emulators to Theory" where you introduce what an emulator is meant to do (current 2.2).
- Finally, present "Experiments/Applications" (current 2.3 + 3).

Thank you for the thorough read and thoughtful structural comments. We agree that explicitly separating the theoretical and experimental components of the manuscript makes for a more streamlined reading experience. We have restructured the manuscript as suggested, and will modify the end of the introduction to reflect these changes.

Changes to introduction (final paragraph): Section 2 first presents our theoretical framework, highlighting that the goal of many emulation techniques is to simplify complex climate dynamics into a linear set of modes associated with the Fokker-Planck and Koopman operators. We then apply this framework to identify potential sources of

error within six emulation techniques, analyzing them from both a theoretical and practical perspective (Sect. 2.3). In Sect. 3, we introduce a series of experiments using simplified climate models and forcing scenarios designed to stress test and evaluate each emulator; these experiments include box models and a modified version of the Lorenz 63 system. Section 4 contains experimental results, showing that response functions consistently outperform other emulators across potential high-error scenarios. We conclude by discussing optimal use cases for each emulator, along with implications for ESMs based on our pedagogical model results (Sect. 5).

2. SDE context: As I understood, Eq.1 is a special case derived by Hasselmann under assumptions such as time-scale separation between fast (weather) and slow (climate) variables, stationarity of the fast system, and viewing fast processes as random forcing. A brief explanation of these conditions and the meaning of each term (e.g., the white noise term representing aggregated fast-variable effects) would aid clarity.

We agree, but want to draw a further distinction within the fast processes. The stochastic term can also represent interannual variability in addition to weather. We will add the following description around Eq. 1 to clarify these points.

Changes around Eq. 1: To understand the statistics of the system and how they may change over time, we follow Hasselmann (1976) in modeling the evolution of a single climate variable using a stochastic differential equation (SDE) (Fig. 2, box 1). We assume time-scale separation between slow climate processes (e.g., ocean, cryosphere, land vegetation) and other, faster sources of variability.

In this framework, the climate is regarded as the statistical mean of a process that appears stochastic in individual realizations. We treat variations occurring either on timescales shorter than climate change (such as short-term weather fluctuations and interannual variability) or in different realizations as stationary, stochastic noise. This allows us to parameterize their influence on the statistics of the chaotic system:

$$\frac{\partial w}{\partial t} = \mathcal{N}(w) + F(t) + \varepsilon \xi(t),$$

where w is the climate variable (or set of variables) of interest (e.g. temperature), F is an external forcing (e.g.  $CO_2$ ),  $\mathcal N$  is the operator governing the evolution of that variable (under slow climate processes),  $\xi$  is a white noise term (aggregated fast effects, including weather and interannual variability), and  $\varepsilon$  is the noise standard deviation.

3. **Equation consistency:** In Fig. 2 you write  $\frac{\partial w}{\partial t} = \mathcal{N}(w, F) + \epsilon \xi$  while Eq. 2 is  $\frac{\partial w}{\partial t} = \mathcal{N}(w) + F(t) + \epsilon \xi$ . These are not equivalent if feedback parameters

depend on forcing. I don't quite understand which formulation is used when and why?

Thank you for catching this. Eq. 2 (and the discussion that follows) has the proper formulation and the formulation in Fig. 2 is a typo. We will correct this in an updated version of the figure (see end of document).

- 4. **FDT introduction**: Before directly introducing linear response theory via the FDT (L.219 ff.), an intermediate step would help:
- Define an unperturbed generator  $K_0$  and perturbation,  $\delta K$ , so  $K=K_0+\delta K$
- Then the perturbed expectation value of an observable g is  $\left\langle g\right\rangle_t^{perturbed} = \left\langle g\right\rangle_t^{(0)} + \left.\delta \left\langle g\right\rangle_t + \left.\mathcal{O}(\delta^2).$
- From there, Ruelle's response theory provides the general solution, with FDT as the special equilibrium case (Eq. 12)

We agree that this helps connect the FDT to other topics in the manuscript, and have added this intermediate step prior to introducing the FDT.

Changes to (newly labeled) Sect. 2.2: To make the relationship between response theory and the Koopman operator explicit in the context of emulation, we first consider the system's dynamics to be governed by an operator, K. When the system is subject to a small external perturbation, this operator can be split into an unperturbed component,  $K_0$ , and the perturbation itself,  $\delta K$ , such that  $K = K_0 + \delta K$ . The expectation value of a statistical quantity g under the perturbed dynamics can be approximated to first order as the sum of its unperturbed evolution,  $\langle g \rangle_0$ , and a linear correction,  $\delta \langle g \rangle$ .

A general solution for this linear correction is provided by Ruelle's response theory. For systems in a statistical steady state (i.e., at equilibrium), this framework simplifies to the Fluctuation Dissipation Theorem (FDT) (Lucarini et al., 2025). The FDT describes how a system (e.g. the Earth system) responds to perturbations (anthropogenic  $CO_2$  emissions) relative to some baseline state (preindustrial conditions). The change in the ensemble average field,  $\delta(g)$ , is obtained by convolving a forcing, F(t), with the system's response function, R(t)

$$\delta\langle g\rangle = \int_{-\infty}^{t} R(s)F(t-s)ds.$$

Formally, the response function is calculated by computing the temporal autocorrelation between the statistical quantity g and the system's score function, s,

$$R(t) = \langle g(t' = t)s(t' = 0) \rangle,$$

where the score function of the steady-state distribution encodes how a small perturbation alters the system's statistics; see Giorgini et al. (2024, 2025b) for more details. The connection to Koopman operator theory is that temporal autocorrelations are expressed explicitly in terms of the Koopman operator, see Zagli et al. (2024).

**Connecting emulators to the theoretical groundwork**

Earth system models generate data that implicitly obeys Eq. 1 (or its operator-based equivalents), but they do not provide us with the exact operators or their solutions. Instead, we observe samples, and the emulator's task is to approximate either the full distribution or selected observables from these samples. The key conceptual contribution of the paper is to connect emulator-based approaches to operator and response formulations. Modern probabilistic models such as Bayesian inference or diffusion models are naturally connected to the state-based view (Fokker–Planck). For example, diffusion models learn a score function (the gradient of the log density), which directly appears in the Fokker–Planck operator. From this perspective, such emulators can be seen as approximating the Fokker-Planck operator, and thus as providing an entry point to linear response theory: score-based response functions are essentially obtained by applying linear response but replacing the Fokker–Planck operator with the learned score. Similarly, emulators that target specific observables (e.g., the mean of the distribution) connect to the Koopman operator and can therefore be framed within linear response theory as well. The authors explicitly work out these connections for six emulation techniques, focusing on temperature anomalies, and discuss the errors each emulator makes in approximating the underlying operators.

**Comments / Suggestions:**

**1. Clarity and structure:**

- This conceptual bridge between emulators and operator theory is central to your paper. I recommend making it an independent section that explicitly highlights these links and their implications for emulator errors (see comment on Framing in the previous section)
- An additional table or expanded version of Table 1 would be very helpful. It could
- summarize pros/cons of each approach, e.g. the computational speed of pattern scaling vs. its structural biases, or the expressiveness of score-based emulators vs. challenges in accessibility and training. In addition, the conncetion of Table 1 to Fig. 2 could be strengthened by adding a column called Emulator type (or adding brackets) that show Method I belongs to Pattern Scaling + Extensions; Method II + IV are impulse response emulators and Method V + VI to Operator-Based emulation

- Figure 1 and Appendix A1 are excellent in motivating the error sources and in giving an example of how your framework helps identifying them.

We agree that expanding Table 1 would improve its utility and can also help to clarify its connection to Fig. 2. We will add an additional column to highlight key pros/cons of each approach and clarify which emulator typology each method belongs to (see end of document).

2. **Data expectations**: When first reading Section 2.2, I expected the framework to be applied to CMIP6 data (reinforced by Fig. 1). It only became clear later.

We agree that there is a framing issue around this point. Based on this comment and similar comments from Reviewer #1, we will make changes to the abstract and introduction to clarify our contribution.

Addition to abstract: To support our theoretical contributions, we provide practical implementation details for each technique, along with discussion on the relative utility of these emulation methods. We evaluate emulator performance using simplified climate models, including box models and a modified version of the Lorenz 63 model, across a series of experiments designed to highlight different potential sources of error.

Changes to the final paragraph of the introduction are given on p.2-3.

We will also include the following section as an addition to the discussion, titled Implications for ESMs, to make the connections between our theoretical and experimental results to ESMs explicit.

Implications for ESMs: While the lack of a common conceptual baseline has historically hindered comparisons between emulator classes, our framework takes an important step towards resolving this. Efforts such as ClimateBench, which provide a common training and evaluation benchmark, have been useful to that end (Watson-Parris et al., 2022), but emulator structural differences prevent this framework from being applied to all existing emulation techniques. Additionally, the high computational burden of running scenarios beyond those in the CMIP archive (for training or evaluation), prevents rigorous assessment of emulator capability (e.g., emulating the impact of individual forcings) and generalizability (accuracy beyond ScenarioMIP). Results from experiments such as the Detection and Attribution MIP (DAMIP) and Regional Aerosol MIP (RAMIP) can help fill these gaps (Gillett et al., 2016; Wilcox et al., 2023), but the field of ESM emulation is currently data-constrained. Our theoretical framework provides value in this data-limited setting, as it allows us to evaluate the assumptions present in

many common emulators. Our results illustrate the potential sources of error different emulator structural assumptions invite, giving us tools to assess and improve emulation techniques independently of ESM results. As ESM outputs improve with CMIP7 and beyond, this framework can help ensure emulators are prepared to train on those new results.

Our pedagogical experiments provide a useful tool to isolate and examine individual sources of error when emulating ESMs (Fig. 1). Though our simplified models are limited in that they lack much of the complexity of full-scale ESMs, our experiments highlight that emulator errors can be proactively resolved through structural changes, regardless of the parent model. For example, our results further support the growing body of literature on the utility of response functions (Freese et al., 2024; Womack et al., 2025; Winkler and Sierra, 2025). Response functions offer improvements over traditional pattern scaling, particularly when considering memory effects in decision-relevant scenarios. They may also emulate longer (post-2100) scenarios by accounting for regional pattern shifts, though longer ESM runs, such as the extensions proposed in ScenarioMIP for CMIP7, are required to test this (Van Vuuren et al., 2025). Existing emulators of ESMs may also benefit from incorporating response functions, c.f., recent work into hybrid emulation using generative machine learning methods in addition to pattern scaling (Bouabid et al., 2025).

Several promising emulation techniques explored here, including the Fluctuation Dissipation Theorem (FDT), Dynamic Mode Decomposition (DMD), and Extended DMD (EDMD), have seen uses in climate science but have yet to be applied directly as emulators of ESM outputs as defined by Tebaldi et al. (2025). An intermediate step for either the FDT or EDMD and DMD may be to first emulate an EMIC, helping determine useful training scenarios without the cost of a full ESM. Our results suggest further research into these techniques is warranted, as they may represent more complex dynamics than other methods. In this context, the FDT stands apart as the most promising technique for emulating general dynamical systems, as evidenced by its skill in this and other recent work (Giorgini et al., 2025b). However, using the FDT to derive response functions through perturbations requires a full initial condition ensemble for every perturbed grid cell/region (Lucarini et al., 2017; Lembo et al., 2020), similar to the Green's Function MIP (Bloch-Johnson et al., 2024), and is likely prohibitively expensive for full ESMs. The score-based FDT (Sect. 2.3) provides a remedy, using statistical learning methods to learn the score function and thus the system response (Giorgini et al., 2025b). Regardless of the derivation method, our results suggest response functions are the dominant emulation technique both in terms of accuracy and interpretability.

Most work studying climate emulation focuses on developing and implementing new approaches in an application-specific manner. Our results show the utility of an operator-based framework for systematic analysis and comparison of climate emulation techniques. The main benefit of this framework is providing a toolkit for understanding trade-offs between emulator complexity and performance while connecting emulation techniques to fundamental principles of statistical mechanics and stochastic systems. We find that memory effects, internal variability, hidden variables, and nonlinearities are potential error sources, and that response function-based emulators consistently outperform other methods, such as pattern scaling and DMD, across all experiments. Emulator performance varies by experimental setup, particularly through the choice of training data, and further work is required to fully characterize these effects. This framework currently relies on simple experiments, and further work is needed to determine if operator-based methods like EDMD can be practically realized to emulate nonlinear processes in full-scale climate models. Our analysis also highlights the FDT's potential for deriving robust, physically-interpretable response functions, though its computational cost is a potential barrier. As interpretability is an ongoing discussion in the emulator community, investing resources in physically-grounded methods like the FDT may go a long way towards increasing the utility of emulators not just for emulation, but for linear system analysis.

**3. Assumptions in pattern scaling vs. impulse response vs. operator-based modelling:**

- In L. 268 & 299 ff: you righty point out that pattern scaling assumes time-invariance and quasi equilibrium and then mention equilibrium conditions in L.
   314 ff. again in terms of FDT. I found the mentioning of two types of equlibrium conditions a bit fuzzy upon first reading and I think it wuld make sense to be a bit more explicit
- In L. 440 ff. you introduce operator-bsed emualtors as the most general class. This makes sense because the previously introduced emulators have some equilibrium assumptions. I feel like the generalisability of this emulator-based framework could be highlighted a bit more; consider making the assumptions in Fig. 2 (arrow from 3b to 3d) more explicit

On the equilibrium condition of pattern scaling, we agree that this can be more explicit and will make the following changes.

Changes to final paragraph of 2.3.1 Pattern scaling and its immediate extensions: In Appendix A1 we show that pattern scaling has two irreducible sources of error when trained on a ScenarioMIP-like forcing: (1) an equilibrium term, where pattern scaling converges to the wrong steady-state value when forcing plateaus and (2) a memory

term, where pattern scaling breaks down when the system responds slowly compared to changes in the forcing. The former stems from the mismatch between training pattern scaling in a transient regime and attempting to use it to project an equilibrium condition. The latter cannot be accounted for within the pattern scaling framework, motivating the need for methods that explicitly capture memory.

On the generalizability of operator-based emulators, we agree, and will add the following discussion to address this and the limitations of data-driven methods to approximate these operators.

Addition to discussion: While EDMD and DMD attempt to approximate the Koopman operator, they are simplified representations and in many cases do not closely approximate the true operator. Despite this, the Koopman and Fokker-Planck operators provide the most useful theoretical basis as they offer a way to directly link disparate forms of emulators. These techniques have the potential to be highly generalizable to scenarios beyond the training data as they can reproduce the system's true dynamics, but further research is required to determine the potential of using operator-based methods directly for climate emulation.

We will also include a legend in Fig. 2 to help clarify assumptions (see end of document).

4. Conclusions: Your reflections in L.838 ff. are supper fitting. For me, the theoretical contributions were the most compelling part of the paper, since many existing studies implement emulators more naively. As you argue, the lack of a conceptual baseline makes it hard to integrate insights across disciplines, and I would encourage you to highlight this contribution more strongly throughout the manuscript.

We are glad the value of a shared conceptual framework resonated with the reviewer. We also agree this contribution can be more explicit, and will emphasize this point.

From paragraph one of Implications for ESMs: Our theoretical framework provides value in this data-limited setting, as it allows us to evaluate the assumptions present in many common emulators. Our results illustrate the potential sources of error different emulator structural assumptions invite, giving us tools to assess and improve emulation techniques independently of ESM results. As ESM outputs improve with CMIP7 and beyond, this framework can help ensure emulators are prepared to train on those new results.

**Experimental approach (sources of error)**

The authors employ four simple climate models (a two-box model, a three-box model, a noisy box model, and a cubic Lorenz system) and drive each with four structurally distinct forcing pathways (abrupt, transient high-emissions, plateau, and overshoot). For each experiment, they train their emulators on data from one scenario and test them against all other scenarios, thereby comparing performance across settings. The experimental design is deliberately simple: it targets errors arising from dynamical features such as memory, hidden variables, noise, and nonlinearities, rather than errors linked to spatial heterogeneity. This choice has clear advantages—the box-model experiments are transparent, reproducible, and well-suited for stress-testing emulator failure modes—but it also carries disadvantages, as spatial patterns are not represented and emulator performance is reduced to a single aggregated metric across boxes.

**Comments / Suggestions:**

1. Data expectations: Reiterating the previous point. Initially, I found the experimental set-up somewhat confusing, as I expected the tests to involve spatially resolved ESM data. The title, abstract, and introduction explicitly mention ESMs; Fig. 1 also presents spatially resolved data; and in Section 2.2 the term "spatial" led me to expect an evaluation of spatial error. In practice, however, the experiments are based on box models with at most three degrees of "spatial" resolution. Applying your framework to toy models is, in my view, valuable—as it provides a controlled setting for isolating and examining specific effects—but I think this would be clearer if explicitly framed as a proof-of-concept.

In light of this and comments from Reviewer #1, we agree that we need to reframe our contribution and clarify the role the simplified climate models play while addressing their limitations in representing more complex climate processes. We will make changes throughout to state this explicitly and clarify that our work focuses on simplified problems, such as the previously stated changes to the abstract and introduction.

From paragraph two of Implications for ESMs: Our pedagogical experiments provide a useful tool to isolate and examine individual sources of error when emulating ESMs (Fig. 1). Though our simplified models are limited in that they lack much of the complexity of full-scale ESMs, our experiments highlight that emulator errors can be proactively resolved through structural changes, regardless of the parent model.

2. **Averaging of results**: The data only ever shows a single evaluation score, while sometimes the models have multiple boxes. Do you average across boxes?

Yes, we average this score across boxes as relative performance across boxes was consistent for all cases analyzed. We will add a comment to clarify this.

Addition to the introductory paragraph of Results section: In the case of models with multiple regions (boxes), we present only a single evaluation score as relative performance across boxes was consistent for all cases analyzed.

- 3. Table 6 summarizes the experimental findings well.
- 4. I appreciated the conclusions in L.816 ff.

**Other minor suggestions**

- Fig. 2:
  - Add a reference to Table 1 to the description (helpful for understanding the boxes on the right given you refer to Fig.2 already in L.77, but mention the Methods I-VI only from L.115 onwards)

We agree, and will add a reference to Table 1 in the description of Fig. 2.

• What is the difference between solid and dashed arrows (e.g., going from 2b to 3b as opposed to 3b to 3d)?

The solid arrows indicate a direct step or result, whereas the dashed arrows indicate a theoretical connection. We have added a legend to Fig. 2 to help clarify this along with the color-coding in the figure (updated Fig. 2 included at the end of this document).

- L. 759: Non-linearities as opposed to nonlinearities throughout the remainder of the manuscript

This is an artifact of LaTeX line breaks, and we will modify the document to ensure this line break doesn't occur.

**Updated Fig. 2:**

**Updated Table 1:**

**Table 1.** Summary of emulation techniques discussed in this work including a short description and their key assumptions; a conceptual overview of these methods can be found in Fig. 2. Fluctuation Dissipation Theorem assumptions are shared with deconvolution and modal fitting emulation techniques. All techniques except the Fluctuation Dissipation Theorem additionally assume no hidden variables.

| Technique                                                                             | Short Description                                                    | Key Assumptions                                                                              | Pros / Cons                                                                                        |
|---------------------------------------------------------------------------------------|----------------------------------------------------------------------|----------------------------------------------------------------------------------------------|----------------------------------------------------------------------------------------------------|
| Method I: Pattern Scaling
(Pattern Scaling and its Immediate Extensions)           | Time-invariant pattern based on global mean temperature              | Climate is always near equilib-
rium; response is instantaneous;
fixed spatial pattern | Computationally efficient /
Structurally biased with ir-
reducible errors                    |
| Method II: Fluctuation Dissipation Theorem (Dynamical System/Impulse Response Theory) | Response functions derived through perturbation ensemble experiments | Perturbations are small; data come from linear response regime                               | Gives interpretable physical response / Requires nonstandard, computationally expensive scenarios  |
| Method III: Deconvolution
(Dynamical System/Impulse
Response Theory)            | Response functions solved for from any general experiment            | Quasi-equilibrium initial condition; influence of noise is small                             | Applicable to any scenario /
Sensitive to noise, can give
non-physical responses             |
| Method IV: Modal Fitting (Dynamical System/Impulse Response Theory)                   | Response functions fit from any general experiment                   | Response is a decaying exponential; few significant modes                                    | Applicable to any scenario / Requires initial guess, can give non-physical responses               |
| Method V: Dynamic Mode
Decomposition (DMD)
(Operator-based Emulation)           | Approximating system dynamics with a linear operator                 | Dynamics are approx. linear;
training data capture relevant dy-
namics                 | Gives interpretable spa-
tiotemporal information
/ Strong assumption of
linearity         |
| Method VI: Extended DMD (Operator-based Emulation)                                    | Approximating system dynamics with nonlinear basis functions         | Basis functions span Koopman operator; dynamics are approx. linear in new basis              | Can theoretically reproduce
any system behavior / Re-
quires selection of basis
functions |